# SAD Neural Networks: Divergent Gradient Flows and Asymptotic Optimality via o-minimal Structures

**Julian Kranz**[a,b]    **Davide Gallon**[c,d]    **Steffen Dereich**[e]    **Arnulf Jentzen**[f,g]

[a]Department of Information Systems,
University of Münster, Germany, email: julian.kranz@uni-muenster.de

[b]Applied Mathematics: Institute for Analysis and Numerics,
University of Münster, Germany, email: julian.kranz@uni-muenster.de

[c]Applied Mathematics: Institute for Analysis and Numerics,
University of Münster, Germany, email: davide.gallon@uni-muenster.de

[d]RiskLab Switzerland, ETH Zürich, Switzerland, email: dgallon@ethz.ch

[e]Applied Mathematics: Institute for Mathematical Stochastics,
University of Münster, Germany, email: steffen.dereich@uni-muenster.de

[f]School of Data Science and School of Artificial Intelligence, The Chinese University
of Hong Kong, Shenzhen (CUHK-Shenzhen), China, email: ajentzen@cuhk.edu.cn

[g]Applied Mathematics: Institute for Analysis and Numerics,
University of Münster, Germany, email: ajentzen@uni-muenster.de

## Abstract

We study gradient flows for loss landscapes of fully connected feedforward neural networks with commonly used continuously differentiable activation functions such as the logistic, hyperbolic tangent, softplus or GELU function. We prove that the gradient flow either converges to a critical point or diverges to infinity while the loss converges to an asymptotic critical value. Moreover, we prove the existence of a threshold $\varepsilon > 0$ such that the loss value of any gradient flow initialized at most $\varepsilon$ above the optimal level converges to it. For polynomial target functions and sufficiently big architecture and data set, we prove that the optimal loss value is zero and can only be realized asymptotically. From this setting, we deduce our main result that any gradient flow with sufficiently good initialization diverges to infinity. Our proof heavily relies on the geometry of o-minimal structures. We confirm these theoretical findings with numerical experiments and extend our investigation to more realistic scenarios, where we observe an analogous behavior.

## 1 Introduction

The success of deep learning in practice has far outpaced our theoretical understanding of neural network training dynamics. While gradient-based optimization methods – such as (stochastic) gradient descent and its variants [77, 75, 43, 35, 57] – routinely achieve near-zero training loss on complex tasks, the theoretical principles governing their convergence remain poorly understood, particularly in realistic, non-convex settings. Even for the mathematically more tractable gradient flow, which can be viewed as the continuous time limit of gradient descent, a fundamental question persists:

Under what conditions does the gradient flow converge to a good local minimum?[1]

---

[1]Note that one cannot always expect convergence to *global* minima [50, 29, 30], but rather *local* minima with low risk level [46, 40, 41].

Recent years have seen significant progress in the theory of deep learning, with convergence analyses often relying on strong assumptions, such as convexity [11, 22], overparameterization [34, 2], specific initialization schemes [48], or Łojasiewicz inequalities [25, 24, 1]. Moreover, most of these works either assume or imply that the iterates of the optimization scheme or the trajectory of the gradient flow stay within a bounded region. At the same time, an increasing body of work shows that this assumption is not always satisfied, with the simplest example being a linear classifier with logistic activation [70, 67, 78, 39].

## 1.1 Contributions

In this work, we bridge the gap outlined above by observing a general **dichotomy**: For a large class of smooth activation functions (such as the logistic, hyperbolic tangent, softplus or GELU function [42] which has been used in BERT [27] and GPT-3 [12]) and loss functions (such as mean squared error or binary cross entropy), the gradient flow either

- converges to a critical point, or
- diverges to infinity with the loss converging to a *generalized critical value* [76].

Using finiteness of the set of generalized critical values [17], we deduce the existence of a threshold $\varepsilon > 0$ such that any gradient flow initialized within $\varepsilon$-distance to the optimal level necessarily converges to it. We refer to Theorem 2.8 for a mathematically rigorous statement of our dichotomy result. We emphasize that convergence of gradient methods to critical points under boundedness assumptions is a well-understood phenomenon [10, 3, 4, 19, 25, 8, 52, 45]. Our main goal in stating the above dichotomy is to make the results from [17] accessible to the deep learning community as the existence of the threshold $\varepsilon > 0$ seems to be unnoticed in the deep learning literature.

As a concrete application of this dichotomy, we complement the divergence results obtained in [70, 67, 78, 39] by a general **divergence result for polynomial target functions**: For a polynomial target function $p \colon \mathbb{R}^n \to \mathbb{R}^m$ of degree at least two, a sufficiently big neural network architecture and a sufficiently big training dataset, we prove that the loss has no global minimum, but infimum zero. In particular, the optimal neural network parameters can only be achieved asymptotically. Thus, by the dichotomy result, the gradient flow diverges whenever the parameters are initialized at a loss level below $\varepsilon$. We refer to Theorem 3.4, Theorem 3.5 and Corollary 3.6 for a mathematically rigorous statement of our divergence result. The "infimum zero" part of our result is essentially well known and a standard ingredient in proofs of the universal approximation theorem [74, 9, 28, 21, 70]. Our novel contribution here is the "no global minimum" part and the resulting divergence phenomenon for the gradient flow.

Our analysis applies to fully connected feedforward deep neural networks with multiple input and output dimensions and a general class of activation functions including the logistic, hyperbolic tangent, softplus and GELU function. Moreover, we can treat both the average loss on a finite dataset and the true loss with respect to a sufficiently smooth probability density. This level of generality arises because our analysis builds on bootstrapping abstract properties rather than concrete computations:

- The functions appearing in our analysis are **definable** in an o-minimal structure in the sense of mathematical model theory [31], which intuitively means that they can be defined using first order statements involving well-behaved operations such as $+, -, \cdot, \div$, exponentials, logarithms, derivatives and anti-derivatives. Definable functions have strong rigidity and finiteness properties such as the uniform finiteness theorem [15], and definable dimension theory [31]. Our proofs use these techniques in a central and novel way.

- We isolate the general class of *Sublinear Analytic Definable* (**SAD**) functions (see Definition 3.1 and Example 3.2) for which our results hold. SAD functions have good permanence properties ensuring that neural networks with SAD activation function are themselves SAD. The main idea of proving the absence of global minima in our divergence result is that the sublinearity of a SAD neural network prevents it from being globally identical to a polynomial of degree at least two, while the analyticity and definability ensure enough rigidity to detect this property on a finite dataset (or compactly supported distribution). On the other hand, analyticity guarantees a sufficient supply of non-trivial Taylor coefficients, allowing polynomials to be approximated arbitrarily well. We believe that our definition of

SAD activation functions provides a convenient theoretical framework for future research on smooth neural networks.

Building on the theoretical framework established earlier, we validate our findings by training neural networks on polynomial target functions using optimization methods such as gradient descent and Adam [57]. The results closely align with the behavior predicted by our gradient flow analysis. We then extend our investigation to more complex and more practically relevant scenarios: numerical solution of PDEs using the Deep Kolmogorov Method [5] and image classification of the MNIST dataset [63]. Across all these experiments, we consistently observe a growth in the norm of the neural network parameters as the loss diminishes. These findings suggest that the divergence phenomenon we analyze actually holds in much broader generality. We include the source code for our numerical simulations at https://github.com/deeplearningmethods/sad.

The paper is structured as follows: We end this introduction by briefly summarizing related work and limitations of ours below. We introduce the necessary mathematical background and formally state the dichotomy result in Section 2. In Section 3, we develop the divergence result for polynomial target functions. Our numerical experiments are reviewed in Section 4. We conclude in Section 5. The proofs of our main results are given in the Appendix.

## 1.2 Related work

The general theory of o-minimal structures originates from [72, 58, 73] and is introduced accessibly in [31, 15, 81]. We specifically use o-minimality of Pfaffian functions [80], Kurdyka's Łojasiewicz inequality [60], definable dimension theory [31], the uniform finiteness theorem [15] and finiteness of the set of generalized critical values [17]. Applications of o-minimal structures to optimization are surveyed in [47]. There is an increasing body of work on the application of o-minimal structures to convergence results for gradient flows and stochastic gradient descent (SGD) under stability assumptions (assuming that the parameters stay within a bounded region) [10, 3, 4, 19, 25, 8, 52, 45]. Conditions ensuring stability are given in [53, 62]. In the unstable regime, there are divergence results which consider homogeneous networks [67, 78], very specific regression problems [39], or use convergence of the loss and absence of global minimizers as an assumption [70]. Other applications of o-minimality to neural networks are given in [55, 56, 54]. Neural network approximation results such as the universal approximation theorem [16, 37, 44, 64] are surveyed in [28, 7]. Our Theorem 3.5 can be extracted from existing literature. The proof in [21] for $\tanh$ for instance generalizes to our setting. Further neural network approximation results are given in [82, 71, 9, 20, 69, 70].

## 1.3 Limitations and outlook

While we prove powerful results for gradient flows, our methods are currently not capable of proving analogous results for (stochastic) gradient descent. Although results exist under stability assumptions (see Subsection 1.2 above), the main limitation in the unstable regime is our lack of control over the Hessian. To mitigate this, one could try to prove that the loss function satisfies a generalized smoothness condition in the sense of [83, 65] (at least along the gradient trajectory) and use this to employ a non-convex convergence proof for SGD. In light of the empirical evidence provided in [83] this approach seems quite promising and will thus be pursued in future work.

Although we do provide empirical evidence for similar results for gradient descent and other optimizers in Section 4, our numerical results are fairly limited by the variety and size of the considered datasets. Since the main focus of our paper are the mathematical theorems, these experiments are intended as a proof of concept rather than an in-depth experimental analysis.

Another limitation of our analysis is that the considered activation functions are required to be at least $C^1$ or, in some cases, analytic. In particular, our analysis does not apply to the ReLU activation function. In fact our divergence result for polynomial target functions does **not** hold for the ReLU function since in this case, global minima exist for shallow networks [49, 26, 23]. Moreover, global minima always exist if one adds an $L^2$-regularization term to the loss function. For the same reason, divergence does not occur if one uses an optimizer with weight decay such as AdamW [66].

In several neural network training scenarios with diverging model parameters, one can prove that the normalized parameters converge, see [51, 78, 59]. This phenomenon known as *directional convergence* is closely related to the *gradient conjecture at infinity* [18] and can be deduced for

$C^2$-functions definable in a *polynomially bounded* o-minimal structure as a consequence of [61].[2] However, the case of o-minimal structures containing the exponential function which is relevant to our setting is still open [18] and remains to be explored.

## 2 A dichotomy for definable gradient flows

In this section, we present our first result (see Theorem 2.8 below) which states that for arbitrary deep neural networks and most of the commonly used loss functions and activation functions (see Example 2.4 below), any gradient flow associated to the training loss landscape either converges to a critical point or diverges to a *generalized critical value* at infinity. The main conceptual reason for this phenomenon is that all the occuring functions are definable in an o-minimal structure in the sense of model theory. We recall the necessary mathematical background below:

**Definition 2.1** (O-minimal structure). A *structure* $\mathcal{S}$ (expanding the real closed field $\mathbb{R}$) is a collection of subsets $\mathcal{S}_n \subseteq \mathcal{P}(\mathbb{R}^n)$ for all $n \in \mathbb{N}$ satisfying the following properties:

   (i) For $A, B \in \mathcal{S}_n$, we have $A \cup B$, $A \cap B$, $\mathbb{R}^n \setminus A \in \mathcal{S}^n$.

   (ii) For $A \in \mathcal{S}_n$ and $B \in \mathcal{S}_m$, we have $A \times B \in \mathcal{S}^{n+m}$.

   (iii) For $A \in \mathcal{S}^n$ and $\pi \colon \mathbb{R}^n \to \mathbb{R}^m$ a coordinate projection with $m \leq n$, we have $\pi(A) \in \mathcal{S}_m$.

   (iv) For each polynomial $p \in \mathbb{R}[X_1, \dots, X_n]$, we have $\{x \in \mathbb{R}^n \mid p(x) > 0\} \in \mathcal{S}^n$ and $\{x \in \mathbb{R}^n \mid p(x) = 0\} \in \mathcal{S}_n$.

A structure $\mathcal{S}$ is called *o-minimal* if all the elements of $\mathcal{S}_1$ are finite unions of points and intervals.

If $\mathcal{S}$ is an o-minimal structure, we call the sets $A \in \mathcal{S}_n$ the *definable* sets. A function $f \colon \mathbb{R}^n \to \mathbb{R}^m$ is called *definable*, if its graph $\{(x, f(x)) \mid x \in \mathbb{R}^n\}$ is definable. If no o-minimal structure is specified, we refer to a set or function as *definable* if it is definable in some o-minimal structure. An instructive (and in fact universal) example of a structure is the one *generated* by a set of functions $\{f_i \colon \mathbb{R}^{n_i} \to \mathbb{R} \mid i \in I\}$ which precisely consists of all sets that can be defined using first order statements involving real numbers and the symbols $+, -, \cdot, \div, <, (f_i)_{i \in I}$.

Standard examples of o-minimal structures are the structure of semialgebraic sets (generated by polynomials), the structure of subanalytic sets (generated by restricted analytic functions) [38], the structure generated by the exponential function [79], and the combination of the latter two [32]. The popular GELU activation function [42] is definable in none of these structures [33, Theorem 5.11], but it is definable in the structure generated by *Pfaffian functions*:

**Definition 2.2** (Pfaffian functions). A $C^1$-function $f \colon \mathbb{R}^n \to \mathbb{R}$ is called *Pfaffian* if there exist $C^1$-functions $f_1, \dots, f_r \colon \mathbb{R}^n \to \mathbb{R}$ with $f_r = f$ such that for any $i = 1, \dots, r$ and $j = 1, \dots, n$, the partial derivative $\frac{\partial f_i}{\partial x_j}(x)$ is a polynomial in $x, f_1(x), \dots, f_i(x)$.

**Theorem 2.3** ([80]). *The structure $\mathbb{R}_{\mathrm{Pfaff}}$ generated by all Pfaffian functions is o-minimal.*

Pfaffian functions include all functions that can be defined iteratively using polynomials, exponentials, logarithms, derivatives and antiderivatives.

**Example 2.4.** The following activation functions $f \colon \mathbb{R} \to \mathbb{R}$ are $C^1$ and definable.

   (i) The *logistic* function $f(x) = \frac{1}{1+e^{-x}}$,

   (ii) The *hyperbolic tangent* function $f(x) = \tanh(x)$,

   (iii) The *softplus* function $f(x) = \log(1 + e^x)$ [36],

   (iv) The *swish* function $f(x) = \frac{x}{1+e^{-\beta x}}$ with $\beta > 0$,

   (v) The *Gaussian error linear unit (GELU)* function $f(x) = x \cdot \frac{1}{2}(1 + \mathrm{erf}(\frac{x}{\sqrt{2}}))$, where $\mathrm{erf}(x) = \frac{2}{\sqrt{\pi}} \int_0^x e^{-t^2} \, dt$ is the Gaussian error function [42],

---

[2]We would like to thank Vincent Grandjean for pointing this out.

(vi) The *Mish* function $f(x) = x \tanh(\log(1 + e^x))$ [68],

(vii) The *exponential linear unit (ELU)* function $f(x) = \begin{cases} e^x - 1, & x \leq 0 \\ x, & x > 0 \end{cases}$ [14],

(viii) The *softsign* function $f(x) = \frac{x}{|x|+1}$.

The following loss functions $\ell \colon \mathbb{R} \times \mathbb{R} \to \mathbb{R}$ are $C^1$ and definable.

(ix) The *squared error* $\ell(x, y) = (x - y)^2$,

(x) The *binary cross-entropy* $\ell(x, y) = -(x \log y + (1 - x) \log(1 - y))$, for $y \in (0, 1)$,

(xi) The *Huber loss* $\ell(x, y) = \begin{cases} \frac{1}{2}(x - y)^2, & |x - y| \leq \delta \\ \delta(|x - y| - \frac{1}{2}\delta), & |x - y| > \delta \end{cases}$ with $\delta \in (0, \infty)$.

The following functions are definable but not $C^1$:

(xiii) The *ReLU* function $f \colon \mathbb{R} \to \mathbb{R}, \quad f(x) = \max(0, x)$,

(xiv) The *absolute error* $\ell \colon \mathbb{R} \times \mathbb{R} \to \mathbb{R}, \quad \ell(x, y) = |x - y|$.

We proceed with a definition of neural network architectures. For a function $\psi \colon \mathbb{R} \to \mathbb{R}$ and an integer $d \in \mathbb{N}$, we denote by $\psi^{(d)} \colon \mathbb{R}^d \to \mathbb{R}^d, \quad \psi^{(d)}(x_1, \dots, x_d) := (\psi(x_1), \dots, \psi(x_d))$ its *amplification*.

**Definition 2.5** (Neural network architectures). A *neural network architecture*[3] is a tuple $\mathcal{A} = (d_0, \dots, d_k, \psi)$ where $d_0, \dots, d_k \in \mathbb{N}$ and $\psi \colon \mathbb{R} \to \mathbb{R}$ is a function called the *activation function*. The *dimension* of $\mathcal{A}$ is given by $d(\mathcal{A}) = \sum_{i=1}^{k} d_i(d_{i-1} + 1)$. Fix an isomorphism $\mathbb{R}^{d(\mathcal{A})} \mathrel{\hat{=}} \prod_{i=1}^{k} \mathbb{R}^{d_i \times d_{i-1}} \times \mathbb{R}^{d_i}$. The *response* $\mathcal{N}_\theta^{\mathcal{A}} \colon \mathbb{R}^{d_0} \to \mathbb{R}^{d_k}$ of the architecture $\mathcal{A}$ and parameters $\theta \mathrel{\hat{=}} (W_1, b_1, \dots, W_k, b_k) \in \mathbb{R}^{d(\mathcal{A})}$ with $W_i \in \mathbb{R}^{d_i \times d_{i-1}}$ and $b_i \in \mathbb{R}^{d_i}$ is given by $\mathcal{N}_\theta^{\mathcal{A}}(x) := f_k \circ \dots \circ f_1(x)$ where $f_1, \dots, f_k$ are defined as

$$f_i(x) = \begin{cases} \psi^{(d_i)}(W_i x + b_i), & i < k \\ W_k x + b_k, & i = k. \end{cases}$$

**Definition 2.6** (Generalized critical values). Let $f \colon \mathbb{R}^d \to \mathbb{R}$ be a differentiable function.

(i) The set of *critical points* of $f$ is the set $\{\theta \in \mathbb{R}^d \mid \nabla f(\theta) = 0\}$.

(ii) The set of *critical values* of $f$ is the set $K_0(f) = \{f(\theta) \in \mathbb{R} \mid \theta \in \mathbb{R}^d, \nabla f(\theta) = 0\}$.

(iii) The set of *asymptotic critical values* of $f$ is the set

$$K_\infty(f) = \{c \in \mathbb{R} \mid \exists (\theta_n)_{n \in \mathbb{N}} \subseteq \mathbb{R}^d \text{ s. t. } \|\theta_n\| \to \infty, \, f(\theta_n) \to c, \, \|\theta_n\| \|\nabla f(\theta_n)\| \to 0\}.$$

(iv) The set of *generalized critical values* of $f$ is the set $K(f) = K_0(f) \cup K_\infty(f)$.

**Definition 2.7** (Locally Lipschitz derivative). A $C^1$-function $f \colon \mathbb{R}^n \to \mathbb{R}^m$ is said to have a *locally Lipschitz derivative*, if for each $x \in \mathbb{R}^n$, there is an open neighbourhood $U \subseteq \mathbb{R}^n$ of $x$ and a constant $L > 0$ such that for all $x_1, x_2 \in U$, we have $\|Df(x_1) - Df(x_2)\| \leq L\|x_1 - x_2\|$.

Below, we characterize the gradient flows of neural network loss landscapes in the $C^1$ definable setting. Our result is mostly a well-known consequence of the available literature. We consider both the empirical loss with respect to a finite dataset (see item (i) below) and the true loss with respect to a continuous data distribution (see item (ii) below).

For a point $x \in \mathbb{R}^n$, we denote by $\delta_x$ the Dirac measure at $x$. For a measure $\mu$ and a measurable function $p \colon \mathbb{R}^n \to [0, \infty)$, we write $d\mu(x) = p(x)dx$ if $\mu$ is given by $\mu(A) = \int_A p(x)dx$ for all measurable sets $A$. The *support* of a function $p \colon \mathbb{R}^n \to [0, \infty)$ is the set $\overline{\{x \in \mathbb{R}^n \mid p(x) \neq 0\}}$.

---

[3]or more precisely a *fully connected feedforward* neural network

**Theorem 2.8** (Dichotomy for gradient flows). *Let $\mathcal{A} = (d_0, \ldots, d_k, \psi)$ be a neural network architecture with a $C^1$ definable activation function $\psi$ (see Example 2.4 (i) - (xi)). Let $f \colon \mathbb{R}^{d_0} \to \mathbb{R}^{d_k}$ and $\ell \colon \mathbb{R}^{d_k} \times \mathbb{R}^{d_k} \to [0, \infty)$ be $C^1$ definable functions (the target and loss functions). Let $\mu$ be a probability measure on $\mathbb{R}^{d_0}$ with expected loss $\mathcal{L} \colon \mathbb{R}^{d(\mathcal{A})} \to \mathbb{R}$ defined by $\mathcal{L}(\theta) := \mathbb{E}_{x \sim \mu}[\ell(\mathcal{N}_\theta^{\mathcal{A}}(x), f(x))]$, such that*

(i) *$\mu = \frac{1}{n} \sum_{i=1}^n \delta_{x_i}$ for some $n \in \mathbb{N}$ and $x_1, \ldots, x_n \in \mathbb{R}^{d_0}$, or*

(ii) *$d\mu(x) = p(x)dx$ for a $C^1$ function $p \colon \mathbb{R}^{d_0} \to [0, \infty)$ such that $\mathcal{L}$ is definable, or*

*that $\mu$ is a convex combination of cases (i) and (ii). Assume moreover that $\ell$, $f$ and $\psi$ have locally Lipschitz derivatives (see Example 2.4 (i) - (xi)). Then the following hold:*

(iii) *For every $\theta_0 \in \mathbb{R}^{d(\mathcal{A})}$, there exists a unique $C^1$ function $\Theta \colon [0, \infty) \to \mathbb{R}^{d(\mathcal{A})}$ satisfying*
$$\Theta(0) = \theta_0, \quad \Theta'(t) = -\nabla\mathcal{L}(\Theta(t)), \quad \forall t \in [0, \infty).$$

(iv) *For every $\Theta$ as in (iii), exactly one of the following holds:*

    (a) *either $\lim_{t \to \infty} \Theta(t)$ exists and is a critical point of $\mathcal{L}$, or*

    (b) *$\lim_{t \to \infty} \|\Theta(t)\| = \infty$ and $\lim_{t \to \infty} \mathcal{L}(\Theta(t))$ is an asymptotic critical value of $\mathcal{L}$.*

(v) *There exists an $\varepsilon > 0$ such that for every $\Theta$ as in (iii) with $\mathcal{L}(\Theta(t_0)) < \inf_{\theta \in \mathbb{R}^{d(\mathcal{A})}} \mathcal{L}(\theta) + \varepsilon$ for some $t_0 \in [0, \infty)$, it holds that $\lim_{t \to \infty} \mathcal{L}(\Theta(t)) = \inf_{\theta \in \mathbb{R}^{d(\mathcal{A})}} \mathcal{L}(\theta)$.*

*Proof.* The proof of Theorem 2.8 is given on page 27 in Appendix B. $\qquad\square$

**Remark 2.9.** Note that Theorem 2.8 applies to all the activation and loss functions listed in Example 2.4 (i) - (xi). In particular, Theorem 2.8 applies to loss functions $\mathcal{L} \colon \mathbb{R}^d \to \mathbb{R}$ of the form $\mathcal{L}(\theta) := \frac{1}{n} \sum_{i=1}^n (\mathcal{N}_\theta^{\mathcal{A}}(x_i) - y_i)^2$ for pairwise distinct $x_1, \ldots, x_n \in \mathbb{R}^{d_0}$ and $y_1, \ldots, y_n \in \mathbb{R}^{d_k}$.

# 3 Divergence of gradient flows for polynomial target functions

In this section, we state the main result of this paper (see Corollary 3.6) saying that for most of the common activation functions and any non-linear polynomial target function, the gradient flow almost always diverges to infinity while the loss converges to zero. Our conceptual starting point to this result is to isolate the key properties of common activation functions into the following definition.

**Definition 3.1** (SAD activation function). A function $f \colon \mathbb{R}^n \to \mathbb{R}^m$ is called *sublinear analytic definable* (**SAD**) if it satisfies the following properties:

(S) $\limsup_{t \to \infty} \frac{\|f(tx)\|}{t} < \infty$, for all $x \in \mathbb{R}^n$,                                         *(sublinear)*

(A) $f$ is analytic,                                                                       *(analytic)*

(D) $f$ is definable in some o-minimal structure (see Definition 2.1).                 *(definable)*

Practically all activation functions used in Machine Learning applications satisfy conditions (S) and (D) above. Moreover, many of them satisfy condition (A) too.

**Example 3.2** (SAD activation functions). The activation functions (i) - (vi) in Example 2.4 are SAD.

**Definition 3.3** (Loss function). We call a function $\ell \colon \mathbb{R}^n \times \mathbb{R}^n \to [0, \infty)$ a *loss function* if and only if it satisfies for all $x, y \in \mathbb{R}^n$ that $x = y \Leftrightarrow \ell(x, y) = 0$.

The main technical contribution of this work states that on a sufficiently big training dataset (or data distribution), a given SAD neural network can never fit any non-linear polynomial perfectly.

We call a Borel measurable set $A \subseteq \mathbb{R}^n$ *conull* if its complement has Lebesgue measure zero.

**Theorem 3.4** (Non-representability of polynomials). *Let $\mathcal{A} = (d_0, \ldots, d_k, \psi)$ be a neural network architecture with SAD activation function $\psi$ (see Example 2.4 (i) - (vi)). Let $f \colon \mathbb{R}^{d_0} \to \mathbb{R}^{d_k}$ be a polynomial of degree at least $2$. Then there exist an integer $N \in \mathbb{N}$ and a conull dense open subset $\mathcal{D}_N \subseteq (\mathbb{R}^{d_0})^N$ such that the following holds: For any measurable loss function $\ell \colon \mathbb{R}^{d_0} \times \mathbb{R}^{d_0} \to [0, \infty)$ and any probability measure $\mu$ on $\mathbb{R}^{d_0}$ such that*

*(i)* $\mu = \frac{1}{n} \sum_{i=1}^{n} \delta_{x_i}$ *for some* $n \geq N$ *and* $x_1, \ldots, x_n \in \mathbb{R}^{d_0}$ *with* $(x_1, \ldots, x_N) \in \mathcal{D}_N$, *or*

*(ii)* $d\mu(x) = p(x)dx$ *for a measurable function* $p \colon \mathbb{R}^{d_0} \to [0, \infty)$, *or*

*such that* $\mu$ *is a convex combination of (i) and (ii), the expected loss* $\mathcal{L} \colon \mathbb{R}^{d(\mathcal{A})} \to [0, \infty)$ *defined by* $\mathcal{L}(\theta) := \mathbb{E}_{x \sim \mu}[\ell(\mathcal{N}_\theta^{\mathcal{A}}(x), f(x))]$ *satisfies* $\mathcal{L}(\theta) > 0$ *for all* $\theta \in \mathbb{R}^{d(\mathcal{A})}$. *Moreover, in the case* $d_0 = 1$, *the set* $\mathcal{D}_N$ *can be chosen as the set of all tuples* $(x_1, \ldots, x_N)$ *for which* $x_1, \ldots, x_N$ *are pairwise distinct.*

*Proof.* The proof of Theorem 3.4 is given on page 29 in Appendix C. □

While Theorem 3.4 prevents the loss $\mathcal{L}(\theta)$ from being exactly zero, we prove that a fixed sufficiently big architecture $\mathcal{A}$ still achieves arbitrarily good approximations. This is a standard result (see [21, 28, 7]) which we include for convenience of the reader.

**Theorem 3.5** (Neural network approximation for polynomials). *Let* $f \colon \mathbb{R}^m \to \mathbb{R}^r$ *be a polynomial of degree* $n \in \mathbb{N}$. *Let* $\mathcal{A} = (d_0, \ldots, d_k, \psi)$ *be a neural network architecture with* $d_0 = m$ *input neurons,* $d_k = r$ *output neurons and a non-polynomial analytic activation function* $\psi$. *Assume that* $\mathcal{A}$ *is sufficiently big in the sense that*

*(i)* *there is a hidden layer* $0 < i < k$ *of size* $d_i \geq r \left( \binom{n+m}{m} - m \right)$,

*(ii)* *all previous hidden layers have size* $\min(d_0, \ldots, d_{i-1}) \geq d_0$,

*(iii)* *all subsequent hidden layers have size* $\min(d_{i+1}, \ldots, d_k) \geq d_k$.

*Then there exists a sequence of parameters* $(\theta_j)_{j \in \mathbb{N}} \subseteq \mathbb{R}^{d(\mathcal{A})}$ *such that* $(\mathcal{N}_{\theta_j}^{\mathcal{A}})_{j \in \mathbb{N}}$ *converges to* $f$ *uniformly on compact sets. In particular, we have* $\inf_{\theta \in \mathbb{R}^{d(\mathcal{A})}} \mathcal{L}(\theta) = 0$ *where* $\mathcal{L}$ *is as in Theorem 3.4.*

*Proof.* The proof of Theorem 3.5 is given on page 35 in Appendix D. □

We can now phrase our main result. It states that for a sufficiently big architecture, a sufficiently big dataset (or distribution) and a sufficiently good initialization, the gradient flow diverges to infinity while the loss converges to zero.

**Corollary 3.6** (Divergence for polynomial target functions). *Let* $f \colon \mathbb{R}^m \to \mathbb{R}^r$ *be a polynomial of degree* $\deg(f) \geq 2$. *Let* $\mathcal{A} = (d_0, \ldots, d_k, \psi)$ *be a neural network architecture with* $d_0 = m$ *input neurons,* $d_k = r$ *output neurons and a non-polynomial SAD activation function* $\psi$ *(see Example 2.4 (i) - (vi)). Assume that* $\mathcal{A}$ *is sufficiently big in the sense that*

*(i)* *there is a hidden layer* $0 < i < k$ *of size* $d_i \geq r \left( \binom{\deg(f)+m}{m} - m \right)$,

*(ii)* *all previous hidden layers have size* $\min(d_0, \ldots, d_{i-1}) \geq d_0$,

*(iii)* *all subsequent hidden layers have size* $\min(d_{i+1}, \ldots, d_k) \geq d_k$.

*Then there is an* $N \in \mathbb{N}$ *and a conull dense open subset* $\mathcal{D}_N \subseteq (\mathbb{R}^{d_0})^N$ *with the following property: For any* $C^1$ *definable loss function* $\ell \colon \mathbb{R}^{d_0} \times \mathbb{R}^{d_0} \to [0, \infty)$ *and any probability measure* $\mu$ *on* $\mathbb{R}^{d_0}$ *with expected loss* $\mathcal{L} \colon \mathbb{R}^{d(\mathcal{A})} \to [0, \infty)$ *defined by* $\mathcal{L}(\theta) := \mathbb{E}_{x \sim \mu}[\ell(\mathcal{N}_\theta^{\mathcal{A}}(x), f(x))]$, *such that*

*(iv)* $\mu = \frac{1}{n} \sum_{i=1}^{n} \delta_{x_i}$ *for some* $n \geq N$ *and* $x_1, \ldots, x_n \in \mathbb{R}^{d_0}$ *with* $(x_1, \ldots, x_N) \in \mathcal{D}_N$, *or*

*(v)* $d\mu(x) = p(x)dx$ *for a* $C^1$ *function* $p \colon \mathbb{R}^{d_0} \to [0, \infty)$ *such that* $\mathcal{L}$ *is definable, or*

*such that* $\mu$ *is a convex combination of (iv) and (v), there exists an* $\varepsilon > 0$ *such that for every* $C^1$ *function* $\Theta \colon [0, \infty) \to \mathbb{R}^d$ *satisfying* $\mathcal{L}(\Theta(t_0)) < \varepsilon$ *for some* $t_0 \in [0, \infty)$ *and* $\Theta'(t) = -\nabla \mathcal{L}(\Theta(t))$ *for all* $t \in [0, \infty)$, *we have* $\lim_{t \to \infty} \mathcal{L}(\Theta(t)) = 0$ *and* $\lim_{t \to \infty} \|\Theta(t)\| = \infty$.

*Proof.* The corollary is a direct consequence of Theorem 2.8, Theorem 3.4 and Theorem 3.5. □

# 4 Numerical experiments

In this section, we empirically validate the theoretical results established in the previous sections (see Corollary 3.6), demonstrating the divergence phenomena of the neural network parameters while the loss converges to zero. Interestingly, we observe that despite a sufficiently good initialization is required in our theorem, in practice it is not required for this phenomenon to occur. Our analysis begins with experiments on polynomial target functions and is then extended to more complex learning tasks, showing that this phenomenon persists beyond our settings and suggesting broader implications for neural network training. Further implementation details are provided in Appendix E.

While our theoretical results hold in the continuous-time gradient flow regime, practical training is typically performed using discrete-time (stochastic) gradient descent. Since gradient descent can be viewed as an explicit discretization of gradient flow, we expect qualitatively similar behavior, with potential numerical differences arising due to discretization.

The gradient flow dynamics of a differentiable loss function $\mathcal{L} \colon \mathbb{R}^d \to \mathbb{R}$ are defined by the ordinary differential equation

$$\theta(0) = \theta_0, \quad \theta'(t) = -\nabla\mathcal{L}(\theta(t)), \quad \forall t \in [0, \infty),$$

which describes the evolution of parameters under infinitesimal gradient updates. Gradient descent corresponds to an explicit (forward) Euler discretization of this flow:

$$\theta_{k+1} = \theta_k + \eta\theta'(t)|_{t=k\eta} = \theta_k - \eta\nabla\mathcal{L}(\theta_k).$$

Here $\theta_k \in \mathbb{R}^d$ denotes the parameter vector at iteration $k$, and $\eta > 0$ denotes the learning rate.

## 4.1 Validation on polynomial target functions

We train fully connected neural networks with three hidden layers to approximate polynomial target functions of varying input dimension. To exhibit the generality of our findings, we consider five different SAD activation functions (see Definition 3.1) and for each of them we plot averages of the loss and the parameter norm (using exponential moving averages and Monte Carlo averaging over 20 independent random initializations).

To most closely resemble the gradient flow dynamics, we begin our analysis with gradient descent, see Figure 1. The slow divergence of parameters is theoretically justified by the fact that the growth of parameter norms along the gradient flow is $\mathcal{O}(\sqrt{t})$ (see (8) in Appendix B). In our setting, the growth seems to be logarithmic as illustrated in the exponentially rescaled Figure 6 in Appendix E. This is consistent with a similar logarithmic behavior proved in [67, Theorem 4.3].

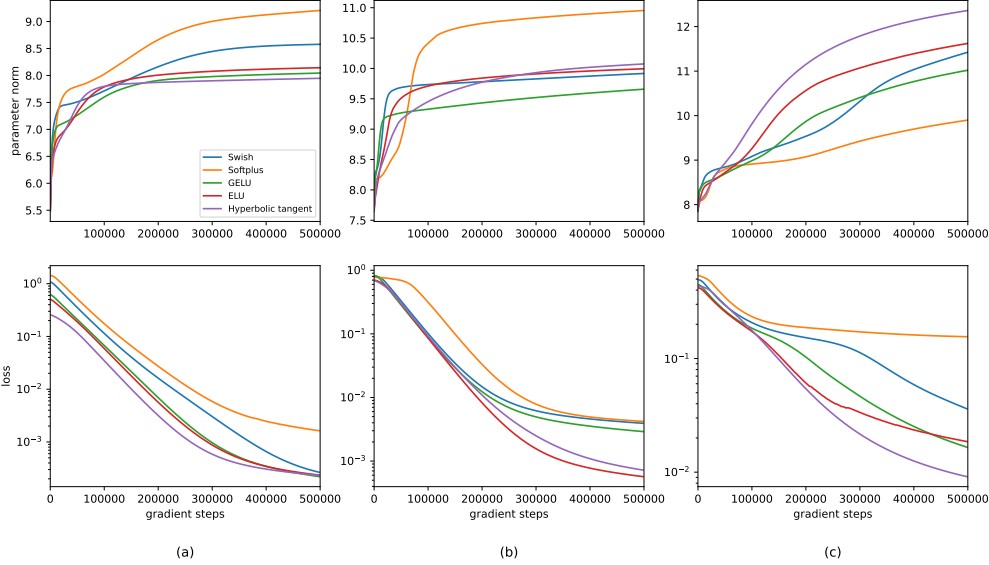

Figure 1: Approximation of polynomial target functions using different activation functions and GD algorithm. From left to right: 1-dimensional input, 2-dimensional input, 4-dimensional input.

Motivated by this limitation, we also employ the Adam optimizer [57], whose adaptive learning rate mitigates the vanishing gradients problem by dynamically rescaling the effective step size. In contrast to gradient descent, where the update magnitude decreases as $\mathcal{O}(1/\sqrt{t})$ with time $t$, Adam maintains substantially larger updates during training, enabling faster convergence in scenarios where gradient descent or SGD would require more iterations to achieve comparable results.

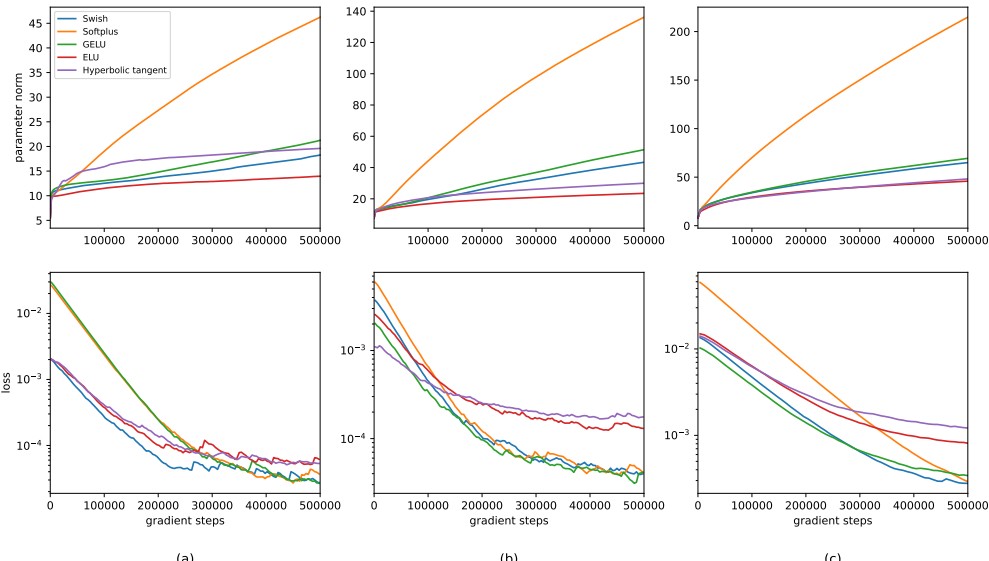

Figure 2: Approximation of polynomial target functions using different activation functions and Adam algorithm. From left to right: 1-dimensional, 2-dimensional, and 4-dimensional input.

In Figure 1 and Figure 2, we consider target functions with 1-, 2-, and 4-dimensional input. In the multi-dimensional cases, we also scale up the network size, increasing the number of hidden neurons from $(d_1, d_2, d_3) = (10, 20, 10)$ to $(20, 40, 20)$. Despite the simplicity of the task and the relatively small network size, we observe a significant growth in the norm of the neural network parameters when the mean squared error loss converges to zero, confirming our theoretical results.

## 4.2 Extension to more complex learning tasks

We investigate whether the divergence behavior observed in the theoretical settings extends to more complex, real-world learning tasks. Interestingly, we find that the same phenomena persist. In particular, the growth of the norm of neural network parameters becomes even more evident in practical scenarios involving high-dimensional data.

In Figure 3 $(a)$, we train deep neural networks to approximate the solutions of two partial differential equations (PDEs): the Heat PDE and the Black–Scholes PDE. These equations are defined as follows:

**Heat equation:**

$$\partial_t u(t, x) = \tfrac{1}{2}\Delta u(t, x), \qquad (t, x) \in [0, T] \times \mathbb{R}^d,$$
$$u(0, x) = \|x\|^2, \qquad x \in \mathbb{R}^d.$$

**Black–Scholes equation:**

$$\partial_t u(t, x) + \sum_{i=1}^{d} \left( (r - c) x_i \partial_{x_i} u(t, x) + \tfrac{1}{2}\sigma_i^2 x_i^2 \partial_{x_i x_i} u(t, x) \right) = 0, \quad (t, x) \in [0, T] \times (0, \infty)^d,$$
$$u(0, x) = e^{-rT} \max\{\max\{x_1, \dots, x_d\} - K, 0\}, \qquad x \in (0, \infty)^d,$$

where $r$ is the risk-free interest rate, $c$ is the cost of carry, $K$ is the strike price and $\sigma$ is the volatility. To solve these PDEs we employ the deep Kolmogorov method [5] which reformulates the PDEs as

stochastic optimization problems using the Feynman–Kac representation. Additionally, in Figure 3 (*b*) we present results from a standard supervised learning task: image classification on the MNIST dataset [63]. This allows us to observe a similar blow-up phenomenon in a classification setting with discrete labels.

In all cases we observe a significant increase in the parameter norm, consistent with the behavior analyzed in the previous section. These results show that our theoretical findings can be generalized to realistic examples and are indicative of a general phenomenon. For the approximation of the PDEs

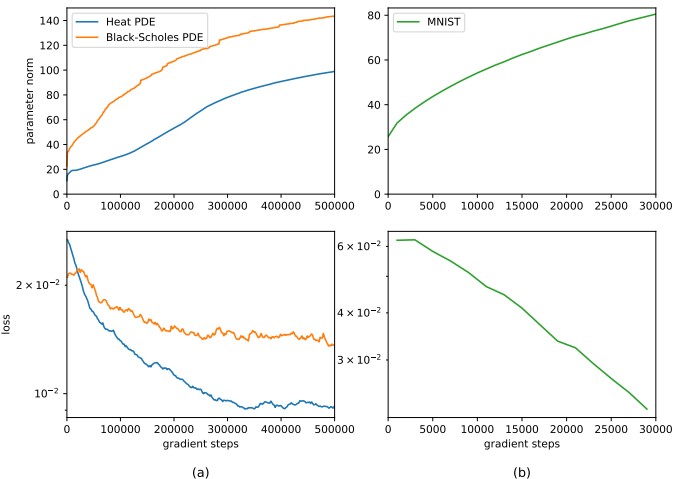

Figure 3: Left: approximation of Heat PDE and Black-Scholes PDE solutions using the deep Kolmogorov method. Right: image classification on the MNIST dataset.

we employ a neural network with three hidden layers and the GELU activation function. The loss function displayed is the relative mean squared error between the model prediction and either the true solution or a Monte Carlo approximation of it. In the MNIST simulations we flatten the input images and train a neural network with GELU activation that gradually decrease the dimension to 10, using the cross-entropy loss to predict the correct input class. To generalize our results, every line represents the exponential moving average of the results, averaged over five independent random initializations.

## 5    Conclusion

We proved that the gradient flow of definable neural networks either converges to a critical point or diverges to infinity – and that the loss values converge to one of finitely many generalized critical values (see Theorem 2.8). We examined the special case of nonlinear polynomial target functions in detail. More specifically, assuming a sufficiently big training dataset and neural network architecture, we showed that if the activation function satisfies a mild condition (see Definition 3.1), then the loss function cannot achieve zero loss (see Theorem 3.4) but it does achieve arbitrarily low values (see Theorem 3.5). From this we deduced our main result stating that in such a situation, the gradient flow diverges to infinity under mild assumptions (see Corollary 3.6). We validated our findings by training neural networks on polynomial target functions using different optimization methods. We then extended our investigation to more realistic scenarios and consistently observed a growth in the norm of the neural network parameters as the loss diminishes, suggesting that the divergence phenomenon we analyzed actually holds in much broader generality (see Section 4).

## Acknowledgements

This work has been supported by the Ministry of Culture and Science NRW as part of the Lamarr Fellow Network. In addition, this work has been partially funded by the Deutsche Forschungsgemeinschaft (DFG, German Research Foundation) under Germany's Excellence Strategy EXC 2044-390685587, Mathematics Münster: Dynamics-Geometry-Structure. This work is partially supported via the AI4Forest project, which is funded by the German Federal Ministry of Education and Research (BMBF; grant number 01IS23025A), and the French National Research Agency (ANR). We gratefully acknowledge the substantial computational resources that were made available to us by the PALMA II cluster at the University of Münster (subsidized by the DFG; INST 211/667-1). The authors are indebted to Floris Vermeulen and Mariana Vicaria for their help in improving the statement of Theorem 3.4. Helpful conversations with Christopher Deninger, Ksenia Fedosova, Allen Gehret, Fabian Gieseke, Vincent Grandjean, and Dennis Wulle are gratefully acknowledged.

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

# Appendix

## A  Locally Lipschitz derivatives

The aim of this section is to establish that the expected loss appearing in Theorem 2.8 has a locally Lipschitz derivative whenever the activation function and the loss function have a locally Lipschitz derivative. This is needed to ensure uniqueness of the gradient flow for the ELU function, the softsign function and the Huber loss in Example 2.4. This section can safely be skipped if one is willing to assume that the activation function and the loss function are $C^2$.

**Lemma A.1.** *The class functions with locally Lipschitz derivatives (see Definition 2.7) is closed under the following operations.*

    (i) *Every $C^2$-function has a locally Lipschitz derivative.*

    (ii) *A function $f = (f_1, \ldots, f_m)\colon \mathbb{R}^n \to \mathbb{R}^m$ has a locally Lipschitz derivative if and only if all of its components $f_1, \ldots, f_m\colon \mathbb{R}^n \to \mathbb{R}$ have locally Lipschitz derivatives.*

    (iii) *If $f\colon \mathbb{R}^{n_1} \to \mathbb{R}$ and $g\colon \mathbb{R}^{n_2} \to \mathbb{R}$ have locally Lipschitz derivatives, then*

$$f \oplus g\colon \mathbb{R}^{n_1+n_2} \to \mathbb{R}, \quad f \oplus g(x, y) = f(x) + g(y)$$

    *and*

$$f \odot g\colon \mathbb{R}^{n_1+n_2} \to \mathbb{R}, \quad f \odot g(x, y) = f(x)g(y)$$

    *have locally Lipschitz derivatives.*

    (iv) *If $f\colon \mathbb{R}^m \to \mathbb{R}^k$ and $g\colon \mathbb{R}^n \to \mathbb{R}^m$ have locally Lipschitz derivatives, then*

$$f \circ g\colon \mathbb{R}^n \to \mathbb{R}^k, \quad f \circ g(x) = f(g(x))$$

    *has a locally Lipschitz derivative.*

    (v) *Let $a, b \in \mathbb{R}$ with $a < b$ and suppose that $f\colon \mathbb{R}^n \to \mathbb{R}$ has a locally Lipschitz derivative. Then the map*

$$F\colon \mathbb{R}^{n-1} \to \mathbb{R}, \quad F(x) = \int_a^b f(t, x)dt$$

    *has a locally Lipschitz derivative.*

*Proof.*    (i)  This is a well-known consequence of the fundamental theorem of calculus.

    (ii) This follows from the inequality

$$\max_{1 \leq i \leq m} \|Df_i(x)\| \leq \|Df(x)\| \leq \sqrt{m} \max_{1 \leq i \leq m} \|Df_i(x)\|$$

    for all $x \in \mathbb{R}^n$.

    (iii) We only prove that $f \odot g$ has a locally Lipschitz derivative since the proof for $f \oplus g$ is similar. Fix $(x_0, y_0) \in \mathbb{R}^{n_1} \times \mathbb{R}^{n_2} = \mathbb{R}^{n_1+n_2}$. Fix $L_f, L_g > 0$ and relatively compact open neighborhoods $x_0 \in U_f \subseteq \mathbb{R}^{n_1}$ and $y_0 \in U_g \subseteq \mathbb{R}^{n_2}$ satisfying

$$\|Df(x_1) - Df(x_2)\| \leq L_f\|x_1 - x_2\|, \quad \forall x_1, x_2 \in U_f$$

    and

$$\|Dg(y_1) - Dg(y_2)\| \leq L_g\|y_1 - y_2\|, \quad \forall y_1, y_2 \in U_g.$$

    Since $f$ and $g$ are $C^1$ and thus locally Lipschitz, there are constants $L'_f, L'_g > 0$ such that

$$\|f(x_1) - f(x_2)\| \leq L'_f\|x_1 - x_2\|, \quad \forall x_1, x_2 \in U_f$$

    and

$$\|g(y_1) - g(y_2)\| \leq L'_g\|y_1 - y_2\|, \quad \forall y_1, y_2 \in U_g.$$

    By continuity and relative compactness, the quantities

$$K_f := \sup_{x \in U_f} |D_f(x)|, \quad K'_f := \sup_{x \in U_f} |f(x)|, \quad K_g := \sup_{y \in U_g} |D_g(y)|, \quad K'_g := \sup_{y \in U_g} |g(y)|$$

are all finite. Defining

$$L := K_f L'_g + K'_f L_g + K'_g L_f + K_g L'_f < \infty,$$

we have for all $(x_1, y_1), (x_2, y_2) \in U_f \times U_g$ that

$$
\begin{aligned}
&\|D(f \odot g)(x_1, y_1) - D(f \odot g)(x_2, y_2)\| \\
&= \left\| \begin{pmatrix} Df(x_1)g(y_1) \\ f(x_1)Dg(y_1) \end{pmatrix} - \begin{pmatrix} Df(x_2)g(y_2) \\ f(x_2)Dg(y_2) \end{pmatrix} \right\| \\
&\leq \|Df(x_1)g(y_1) - Df(x_2)g(y_2)\| + \|f(x_1)Dg(y_1) - f(x_2)Dg(y_2)\| \\
&\leq \|Df(x_1)\|\|g(y_1) - g(y_2)\| + \|Df(x_1) - Df(x_2)\|\|g(y_2)\| \\
&\quad + \|f(x_1) - f(x_2)\|\|Dg(y_1)\| + \|f(x_2)\|\|Dg(y_1) - Dg(y_2)\| \\
&\leq (K_f L'_g + K'_f L_g)\|y_1 - y_2\| + (K'_g L_f + K_g L'_f)\|x_1 - x_2\| \\
&\leq L\|(x_1, y_1) - (x_2, y_2)\|.
\end{aligned}
$$

This proves that $f \odot g$ has a locally Lipschitz derivative.

(iv) Fix $x_0 \in \mathbb{R}^n$. By assumption, there are $L_f, L_g > 0$ and relatively compact open neighbourhoods $g(x_0) \in U_f \subseteq \mathbb{R}^m$ and $x_0 \in U_g \subseteq \mathbb{R}^n$ satisfying

$$\|Df(y_1) - Df(y_2)\| \leq L_f\|y_1 - y_2\|, \quad \forall y_1, y_2 \in U_f$$

and

$$\|Dg(x_1) - Dg(x_2)\| \leq L_g\|x_1 - x_2\|, \quad \forall x_1, x_2 \in U_g.$$

By continuity of $g$, we may assume that $g(U_g) \subseteq U_f$. Since $g$ is $C^1$ and thus itself locally Lipschitz, we may moreover assume that there is $L'_g > 0$ such that

$$\|g(x_1) - g(x_2)\| \leq L'_g\|x_1 - x_2\|, \quad \forall x_1, x_2 \in U_g.$$

By continuity of $Df$ and $Dg$ and relative compactness of $U_f$ and $U_g$, we have

$$L := L_f L'_g \sup_{x \in U_g} \|Dg(x)\| + L_g \sup_{y \in U_f} \|Df(y)\| < \infty.$$

Moreover, for all $x_1, x_2 \in U_f$, we have

$$
\begin{aligned}
\|D(f \circ g)(x_1) - D(f \circ g)(x_2)\| &= \|Df(g(x_1))Dg(x_1) - Df(g(x_2))Dg(x_2)\| \\
&\leq \|Df(g(x_1)) - Df(g(x_2))\|\|Dg(x_1)\| \\
&\quad + \|Df(g(x_2))\|\|Dg(x_1) - Dg(x_2)\| \\
&\leq L\|x_1 - x_2\|.
\end{aligned}
$$

This proves that $f \circ g$ has a locally Lipschitz derivative.

(v) Fix $x_0 \in \mathbb{R}^n$. Since $f$ has a locally Lipschitz derivative, there is for every $t \in [a, b]$ a constant $L_t > 0$ and an open neighbourhood $(t, x_0) \in U_t \subseteq \mathbb{R}^n$ such that for all $(t_1, x_1), (t_2, x_2) \in U_t$, we have

$$\|Df(t_1, x_1) - Df(t_2, x_2)\| < L_t\|(t_1, x_1) - (t_2, x_2)\|.$$

By compactness, we can find $t_1, \ldots, t_k \in [a, b]$ such that $[a, b] \times \{x_0\} \subseteq \bigcup_{i=1}^{k} U_{t_i}$. Without loss of generality, we may assume that each $U_{t_i}$ for $i = 1, \ldots, k$ is of the form $(a_i, b_i) \times V$ for an open neighbourhood $x_0 \in V \subseteq \mathbb{R}^{n-1}$ and $a_i, b_i \in \mathbb{R}$. Now define $L := (b - a) \max_{1 \leq i \leq k} L_{t_i}$ and let $x_1, x_2 \in V$ be arbitrary. Using the Leibniz integral rule, we get

$$
\begin{aligned}
\|DF(x_1) - DF(x_2)\| &= \left\| \int_a^b (D_x f(t, x_1) - D_x f(t, x_2))dt \right\| \\
&\leq \int_a^b \|Df(t, x_1) - Df(t, x_2)\|dt \\
&\leq \int_a^b \max_{1 \leq i \leq k} L_{t_i}\|x_1 - x_2\|dt \\
&\leq L\|x_1 - x_2\|,
\end{aligned}
$$

where $D_x f(t, x_i)$ denotes the differential of the map $x \mapsto f(t, x)$ at $x = x_i$. This proves that $F$ has a locally Lipschitz derivative. $\qquad \square$

**Corollary A.2.** *Let $\mathcal{A} = (d_0, \ldots, d_k, \psi)$ be a neural network architecture with a $C^1$ activation function $\psi \colon \mathbb{R} \to \mathbb{R}$ with a locally Lipschitz derivative. Let $\ell \colon \mathbb{R}^{d_k} \times \mathbb{R}^{d_k} \to [0, \infty)$ be a $C^1$ loss function with a locally Lipschitz derivative. Let $f \colon \mathbb{R}^{d_0} \to \mathbb{R}^{d_k}$ be a $C^1$ target function with a locally Lipschitz derivative. Let $\mu$ be a probability measure on $\mathbb{R}^{d_0}$ such that either*

   *(i) $\mu = \frac{1}{n} \sum_{i=1}^{n} \delta_{x_i}$ for some $n \in \mathbb{N}$ and $x_1, \ldots, x_n \in \mathbb{R}^{d_0}$, or*

   *(ii) $d\mu(x) = p(x)dx$ for a $C^1$ function $p \colon \mathbb{R}^{d_0} \to [0, \infty)$ of compact support.*

*Then the function*
$$\mathcal{L} \colon \mathbb{R}^{d(\mathcal{A})} \to \mathbb{R}, \quad \mathcal{L}(\theta) = \mathbb{E}_{x \sim \mu}[\ell(\mathcal{N}_\theta^{\mathcal{A}}(x), f(x))]$$
*has a locally Lipschitz derivative.*

*Proof.* This immediately follows from Lemma A.1 since $\mathcal{L}$ can be constructed iteratively from $\psi$ and linear functions using the operations (i) - (v). $\qquad \square$

# B  Proof of Theorem 2.8

This section gives a proof of Theorem 2.8 which is probably well-known to experts and follows standard techniques. The proof is mostly given for lack of a precise reference and for convenience of the reader. Before giving the technical details, we outline the main ideas below.

**Idea of proof**

Existence and uniqueness of the gradient flow for $\mathcal{L}$ follow from the fact that $\mathcal{L}$ is $C^1$ and has a locally Lipschitz derivative. The latter assumption is checked in Appendix A. Now one can distinguish two cases: Either the gradient flow visits some bounded set infinitely often or it does not. In the first case, we use Kurdyka's definable Łojasiewicz inequality [60] and follow an argument from [1, Theorem 2.2] to prove convergence to a critical point (see Lemma B.1). In the second case, the parameters diverge by assumption. We check that the limit is a generalized critical value using the fact that $\mathcal{L}$ is bounded below and therefore has square-integrable gradients along any gradient flow trajectory (see Lemma B.2).

For a point $x_0 \in \mathbb{R}^n$ and $R > 0$, we denote by
$$B_R(x_0) \coloneqq \{x \in \mathbb{R}^n \mid \|x - x_0\| < R\} \subseteq \mathbb{R}^n$$
the open ball of radius $R$ centered at $x_0$.

**Lemma B.1.** *Let $\mathcal{L} \colon \mathbb{R}^d \to [0, \infty)$ be a $C^1$ definable function and let $\Theta \colon [0, \infty) \to \mathbb{R}^d$ a $C^1$-function satisfying the differential equation $\Theta'(t) = -\nabla \mathcal{L}(\Theta(t))$ for all $t \in [0, \infty)$. Suppose that there is a point $\theta_* \in \mathbb{R}^d$ satisfying*
$$\liminf_{t \to \infty} \|\Theta(t) - \theta_*\| = 0. \tag{1}$$
*Then we have $\lim_{t \to \infty} \Theta(t) = \theta_*$.*

*Proof.* Without loss of generality, we may assume that $\mathcal{L}(\theta_*) = 0$. Note that the function $\mathcal{L} \circ \Theta \colon [0, \infty) \to \mathbb{R}$ is non-increasing since we have $\frac{d}{dt} \mathcal{L}(\Theta(t)) = -\|\nabla \mathcal{L}(\Theta(t))\|^2$ for all $t \in [0, \infty)$. Since $\mathcal{L}$ is continuous, (1) and the fact that $\mathcal{L}$ is non-increasing imply that
$$\lim_{t \to \infty} \mathcal{L}(\Theta(t)) = \mathcal{L}(\theta_*) = 0. \tag{2}$$

Now fix $\varepsilon > 0$. By [60, Theorem 2 (b)], there is a continuous strictly increasing definable function $\sigma \colon [0, \infty) \to [0, \infty)$ with $\sigma(0) = 0$ such that whenever $t_1, t_2 \in [0, \infty)$ with $t_1 < t_2$ satisfy $\theta([t_1, t_2]) \subseteq B_{2\varepsilon}(\theta_*)$, we have
$$\int_{t_1}^{t_2} \|\Theta'(t)\| dt \leq \sigma(|\mathcal{L}(\Theta(t_2)) - \mathcal{L}(\Theta(t_1))|). \tag{3}$$

Since $t \mapsto \mathcal{L}(\Theta(t))$ is nonincreasing, (3) implies that

$$\int_{t_1}^{t_2} \|\Theta'(t)\| dt \leq \sigma(|\mathcal{L}(\Theta(t_2)) - \mathcal{L}(\Theta(t_1))|) \leq \sigma(\mathcal{L}(\Theta(t_1))) \tag{4}$$

for all $t_1, t_2 \in [0, \infty)$ satisfying $\Theta([t_1, t_2]) \subseteq B_{2\varepsilon}(\theta_*)$.

By continuity of $\sigma$ and (2), we can find a large enough $t_1 > 0$ such that $\sigma(\mathcal{L}(\Theta(t_1))) < \frac{\varepsilon}{2}$. By (1), we may moreover assume that $\Theta(t_1) \in B_{\frac{\varepsilon}{2}}(\theta_*)$. We claim that we have $\Theta(t) \in B_{\varepsilon}(\theta_*)$ for all $t > t_1$.

Assuming the contrary, there is a smallest $t_2 > t_1$ satisfying $\|\Theta(t_2) - \Theta_*\| = \varepsilon$. But then we have $\Theta([t_1, t_2]) \subseteq B_{2\varepsilon}(\theta_*)$ and by an application of (4), we get

$$\|\Theta(t_2) - \theta_*\| \leq \|\Theta(t_2) - \Theta(t_1)\| + \|\Theta(t_1) - \theta_*\|$$
$$< \int_{t_1}^{t_2} \|\Theta'(t)\| dt + \frac{\varepsilon}{2}$$
$$\leq \sigma(\mathcal{L}(\Theta(t_1))) + \frac{\varepsilon}{2}$$
$$< \varepsilon,$$

a contradiction. Thus, we have $\Theta([t_1, \infty)) \subseteq B_{\varepsilon}(\theta_*)$. Since $\varepsilon > 0$ was arbitrary, the lemma is proven. $\square$

**Lemma B.2.** *Let $\mathcal{L} \colon \mathbb{R}^d \to [0, \infty)$ be a $C^1$ function. Then for any $\theta_0 \in \mathbb{R}^d$, there is a $C^1$-function $\Theta \colon [0, \infty) \to \mathbb{R}^d$ satisfying*

$$\Theta'(t) = -\nabla\mathcal{L}(\Theta(t)), \quad \Theta(0) = \theta_0 \tag{5}$$

*for all $t \in [0, \infty)$. Moreover, the following statements are true:*

    *(i) If $\mathcal{L}$ moreover has a locally Lipschitz derivative, then the solution $\Theta$ to (5) is unique.*

    *(ii) If $\mathcal{L}$ is moreover definable, then exactly one of the following two statements holds true.*

        *(a) $\lim_{t \to \infty} \Theta(t)$ exists and is a critical point of $f$, or*
        *(b) $\lim_{t \to \infty} \|\Theta(t)\| = \infty$ and $\lim_{t \to \infty} \mathcal{L}(\Theta(t))$ is a generalized critical value of $\mathcal{L}$ (see Definition 2.6).*

*Proof.* The existence of a $C^1$-map $\Theta \colon [0, \infty) \to \mathbb{R}^d$ satisfying (5) follows for instance from [54, Proposition 2.11]. The uniqueness part in item (i) follows from the Picard–Lindelöf theorem.

Now assume that $\mathcal{L}$ is moreover definable. We consider two cases:

**Case 1:** We have $\liminf_{t \to \infty} \|\Theta(t)\| < \infty$.

In this case, there is a constant $R > 0$ and a sequence $(t_n)_{n \in \mathbb{N}}$ in $[0, \infty)$ satisfying $\Theta(t_n) \in B_R(0)$ for all $n \in \mathbb{N}$ and $\lim_{n \to \infty} t_n = \infty$. By compactness, a subsequence of $(\Theta(t_n))_{n \in \mathbb{N}}$ converges to some point $\theta_* \in B_R(0)$. In particular, we have $\liminf_{t \to \infty} \|\Theta(t) - \theta_*\| = 0$. Therefore, Lemma B.1 implies that

$$\lim_{t \to \infty} \Theta(t) = \theta_*. \tag{6}$$

To check that $\theta_*$ is a critical point, note that since $\mathcal{L}$ takes only non-negative values, we have

$$\int_0^t \|\Theta'(s)\|^2 ds = -\int_0^t \langle \Theta'(s), \nabla\mathcal{L}(\Theta(s))\rangle ds = \mathcal{L}(\Theta(0)) - \mathcal{L}(\Theta(t)) \leq \mathcal{L}(\Theta(0)) \tag{7}$$

for all $t \in [0, \infty)$. In particular, $t \mapsto \|\Theta'(t)\|$ is square-integrable. In combination with (6), this proves

$$\|\nabla\mathcal{L}(\theta_*)\| = \liminf_{t \to \infty} \|\nabla\mathcal{L}(\Theta(t))\| = \liminf_{t \to \infty} \|\Theta'(t)\| = 0,$$

so that $\theta_*$ is a critical point of $\mathcal{L}$.

**Case 2:** We have $\liminf_{t \to \infty} \|\Theta(t)\| = \infty$.

In this case, the limit $c := \lim_{t\to\infty} \mathcal{L}(\Theta(t))$ exists since the map $t \mapsto \mathcal{L}(\Theta(t))$ is decreasing and bounded below. It remains to be shown that $\liminf_{t\to\infty} \|\Theta(t)\|\|\nabla\mathcal{L}(\Theta(t))\| = 0$. By [39, Proposition 2.54], we have

$$\|\Theta(t)\| \leq \|\Theta(0)\| + \sqrt{t\mathcal{L}(\Theta(0))}, \tag{8}$$

and thus

$$\|\Theta(t)\|\|\nabla\mathcal{L}(\Theta(t))\| = \|\Theta(t)\|\|\Theta'(t)\| \leq \left(\|\Theta(0)\| + \sqrt{t\mathcal{L}(\Theta(0))}\right)\|\Theta'(t)\|, \tag{9}$$

for every $t \in [0, \infty)$.

Assume by contradiction that we have

$$\varepsilon := \liminf_{t\to\infty} \left(\|\Theta(0)\| + \sqrt{t\mathcal{L}(\Theta(0))}\right)\|\Theta'(t)\| > 0.$$

Then there exists $T \in (0, \infty)$ such that $\inf_{t\geq T}\left(\|\Theta(0)\| + \sqrt{t\mathcal{L}(\Theta(0))}\right)\|\Theta'(t)\| > \frac{\varepsilon}{2}$. Thus, we get

$$\infty = \int_T^\infty \left(\|\Theta(0)\| + \sqrt{t\mathcal{L}(\Theta(0))}\right)^{-2} dt \leq \frac{4}{\varepsilon^2}\int_T^\infty \|\Theta'(t)\|^2 dt,$$

contradicting (7). Thus we have $\liminf_{t\to\infty}\left(\|\Theta(0)\| + \sqrt{t\mathcal{L}(\Theta(0))}\right)\|\Theta'(t)\| = 0$. Together with (9), this proves that $\liminf_{t\to\infty} \|\Theta(t)\|\|\nabla\mathcal{L}(\Theta(t))\| = 0$ and concludes the proof of the lemma. $\quad\square$

*Proof of Theorem 2.8.* It is either easy to see or stated as an assumption that $\mathcal{L}$ is definable. Moreover, it is an easy consequence of the Leibniz integral rule that $\mathcal{L}$ is $C^1$. Thus, [17] is applicable and establishes that the set $K(\mathcal{L})$ of generalized critical values of $\mathcal{L}$ (see Definition 2.6) is finite. Moreover, $\mathcal{L}$ has a locally Lipschitz derivative by Corollary A.2. Now the theorem follows from Lemma B.2 if we choose $\varepsilon > 0$ small enough so that the open interval $(\inf_{\theta\in\mathbb{R}^d}\mathcal{L}(\theta), \inf_{\theta\in\mathbb{R}^d}\mathcal{L}(\theta) + \varepsilon)$ does not contain any generalized critical value. $\quad\square$

# C Proof of Theorem 3.4

This section is dedicated to the proof of our main technical contribution Theorem 3.4. This is the main part where o-minimal structures are used in a new way which cannot be found in the neural network literature. We outline the main ideas of the proof below.

**Idea of proof**

The starting point is that the sublinearity condition (S) in Definition 3.1 also implies that the neural network responses $\mathcal{N}_\theta^\mathcal{A}$ are sublinear (see Lemma C.2). In particular, it cannot be equal as a function to a polynomial $f$ of degree at least 2 (see Lemma C.3). This does not explain why $\mathcal{L}$ does not have any zeros yet since $\mathcal{N}_\theta^\mathcal{A}$ and $f$ could be equal on the dataset or data distribution for some $\theta$. The main bulk of the proof now exploits the remarkable rigidity properties of definable analytic functions to show that this cannot happen (see Lemma C.4). We treat the one-dimensional and multi-dimensional input cases separately. In the one-dimensional case, the combination of the identity theorem for analytic functions and the uniform finiteness theorem from o-minimal geometry imply that for any $\theta$, $\mathcal{N}_\theta^\mathcal{A}$ and $f$ cannot be equal on a sufficiently large finite dataset where the required size can be made independent of $\theta$. In the multidimensional case, the conclusion can fail for very degenerate datasets (for example if all datapoints lie on a line along which the polynomial is linear). But by combining the ideas from the one-dimensional case with dimension theory for definable sets, we show that the space of such "bad datasets" has strictly smaller dimension than the space of all datasets. Therefore its complement contains an open dense subset of full Lebesgue measure. The use of definable dimension theory was kindly suggested to us by Floris Vermeulen and Mariana Vicaria.

**Lemma C.1.** *Let $f\colon \mathbb{R} \to \mathbb{R}$ be a $C^1$ definable function. Then $f$ satisfies condition (S) of Definition 3.1 if and only if its derivative is a bounded function.*

*Proof.* Assume first that $C := \sup_{t\in\mathbb{R}}|f'(t)| < \infty$ and fix $x \in \mathbb{R}$. Then the fundamental theorem of calculus implies for any $t \in (0, \infty)$ that

$$\frac{|f(tx)|}{t} \leq \frac{1}{t}\left(|f(0)| + \int_0^1 |f'(stx)||tx|ds\right) \leq \frac{|f(0)|}{t} + C|x| \xrightarrow{t\to\infty} C|x| < \infty,$$

proving condition (S). We prove the converse and assume $f'$ is unbounded. Since $f'$ is definable [15, Lemma 6.1], it is monotone outside a compact interval $[-R, R]$ by the monotonicity theorem [15, Theorem 2.1]. Without loss of generality, we may assume that $f'$ is increasing and unbounded on $[R, \infty)$. The cases that $f$ is decreasing and unbounded on $[R, \infty)$, increasing and unbounded on $(-\infty, R]$, or decreasing and unbounded on $(-\infty, R]$ are analogous. Fix a constant $K > 0$. Since $f'$ is increasing and unbounded on $[R, \infty)$, there is a constant $R' > R$ such that $f'(t) > K$ for all $t > R'$. Thus for all $t \in [R', \infty)$, we have

$$
\begin{aligned}
\limsup_{t \to \infty} \frac{f(t \cdot 1)}{t} &= \limsup_{t \to \infty} \left| \frac{f(R')}{t} + \frac{1}{t} \int_{R'}^{t} f'(s) ds \right| \\
&\geq \limsup_{t \to \infty} \left( \frac{-|f(R')|}{t} + \frac{1}{t} \int_{R'}^{t} f'(s) ds \right) \\
&\geq \limsup_{t \to \infty} \frac{-|f(R')| + (t - R') K}{t} \\
&= K.
\end{aligned}
$$

Since $K > 0$ was arbitrary, we conclude that $\limsup_{t \to \infty} \frac{f(t \cdot 1)}{t} = \infty$, disproving condition (S). $\quad\square$

**Lemma C.2.** *Let $\mathcal{A} = (d_0, \ldots, d_k, \psi)$ be a neural network architecture and $\theta \in \mathbb{R}^{d(\mathcal{A})}$. If the activation $\psi \colon \mathbb{R} \to \mathbb{R}$ is SAD, then the response $\mathcal{N}_\theta^{\mathcal{A}} \colon \mathbb{R}^{d_0} \to \mathbb{R}^{d_k}$ is SAD, too.*

*Proof.* It is easy to see that $\mathcal{N}_\theta^{\mathcal{A}}$ satisfies conditions (A) and (D) of Definition 3.1 for all $\theta \in \mathbb{R}^d$. It remains to verify (S). Since condition (S) can be checked on each component of $\mathcal{N}_\theta^{\mathcal{A}}$ we may assume without loss of generality that $\mathcal{A} = (d_0, \ldots, d_k, \psi)$ has only $d_k = 1$ output neuron.

We prove that $\mathcal{N}_\theta^{\mathcal{A}}$ satisfies condition (S) by induction on $k$. Assume first that $k = 1$. Then $\mathcal{N}_\theta^{\mathcal{A}}$ is of the form

$$
\mathcal{N}_\theta^{\mathcal{A}}(x) = w_1 x_1 + \cdots + w_{d_0} x_{d_0} + b
$$

for $\theta = (w_1, \ldots, w_n, b)$. Then, $\mathcal{N}_\theta^{\mathcal{A}}$ is affine linear and thus satisfies condition (S).

Assume now that we have proven the statement for $k - 1$ and fix $\theta \in \mathbb{R}^d$ and $x \in \mathbb{R}^{d_0}$. We define a smaller architecture $\mathcal{A}' = (d_0, \ldots, d_{k-2}, 1, \psi)$ with only $k - 1$ layers by deleting the $(k-1)$-th layer. Note that by rearranging the components of $\theta \in \mathbb{R}^d$ as $\theta = (\theta_1, \ldots, \theta_{d_{k-1}}, w_1, \ldots, w_{d_{k-1}}, b)$, for appropriate $w_1, \ldots, w_{d_{k-1}}, b \in \mathbb{R}$ and $\theta_1, \ldots, \theta_{d_{k-1}} \in \mathbb{R}^{(d-1)/d_{k-1}-1}$, we can write

$$
\mathcal{N}_\theta^{\mathcal{A}}(tx) = \sum_{i=1}^{d_{k-1}} w_i \psi(\mathcal{N}_{\theta_i}^{\mathcal{A}'}(tx)) + b \tag{10}
$$

for all $t \in \mathbb{R}$. By the induction hypothesis, the functions $t \mapsto \mathcal{N}_{\theta_i}^{\mathcal{A}'}(tx)$ satisfiy condition (S) for every $i = 0, \ldots, d_{k-1}$. In particular, they have bounded derivatives by Lemma C.1. By another application of Lemma C.1 and the chain rule, it follows that the functions $t \mapsto w_i \psi(\mathcal{N}_{\theta_i}^{\mathcal{A}'}(tx))$ for $i = 0, \ldots, d_{k-1}$ satisfy condition (S), too. By linearity and Lemma C.1 applied to (10), it follows that the function $t \mapsto \mathcal{N}_\theta^{\mathcal{A}}(tx)$ satisfies condition (S). Since $x \in \mathbb{R}^{d_0}$ was arbitrary, we conclude that $\mathcal{N}_\theta^{\mathcal{A}}$ satisfies condition (S). $\quad\square$

**Lemma C.3.** *Let $f \colon \mathbb{R}^m \to \mathbb{R}^r$ be a polynomial. Then $f$ is SAD if and only if $f$ is affine linear.*

*Proof.* Note that by definition, polynomials are analytic and definable. Moreover, every linear function $p \colon \mathbb{R}^m \to \mathbb{R}^r$ satisfies condition (S) in Definition 3.1 since we have $\frac{\|p(tx)\|}{t} = \|p(x)\|$ for all $x \in \mathbb{R}^m$ and $t \in (0, \infty)$. Conversely, let $p \colon \mathbb{R}^m \to \mathbb{R}^r$ be polynomial of degree at least $2$ and write $p(x) = (p_1(x), \ldots, p_r(x))$ for $x \in \mathbb{R}^m$ and polynomials $p_1, \ldots, p_r \colon \mathbb{R}^m \to \mathbb{R}$. To prove that $p$ does not satisfy condition (S), it suffices to prove that one of the components $p_1, \ldots, p_r$ does not satisfy condition (S). Thus, we may without loss of generality assume that $r = 1$. We write $p$ as $p(x) = p_H(x) + p_L(x)$ where $p_H$ is a homogeneous polynomial of degree $k \geq 2$ and $p_L$ is a

polynomial of degree strictly less than $k$. Fix $x_0 \in \mathbb{R}^m$ such that $\|x_0\| = 1$ and $p_H(x_0) \neq 0$. Since $p_L$ has degree strictly less than $k$, the limit $\alpha := \lim_{t \to \infty} \frac{|p_L(tx_0)|}{t^{k-1}} \in \mathbb{R}$ exists. We have

$$
\begin{aligned}
\limsup_{t \to \infty} \frac{|p(tx_0)|}{t} &\geq \limsup_{t \to \infty} \frac{|p(tx_0)|}{t^{k-1}} \\
&\geq \limsup_{t \to \infty} \left( \frac{|p_H(tx_0)|}{t^{k-1}} - \frac{|p_L(tx_0)|}{t^{k-1}} \right) \\
&\geq \limsup_{t \to \infty} \frac{t^k |p_H(x_0)|}{t^{k-1}} - \alpha \\
&= \infty.
\end{aligned}
$$

$\square$

The next lemma helps us to detect non-representability of polynomials on a finite dataset. We are indebted to Floris Vermeulen and Mariana Vicaria for explaining to us the proof in the case $m > 1$.

**Lemma C.4.** *Let $f \colon \mathbb{R}^m \times \mathbb{R}^d \to \mathbb{R}$ be a definable analytic function such that for each $\theta \in \mathbb{R}^d$, the function $x \mapsto f(\theta, x)$ is not identically zero. Then there is an integer $N \in \mathbb{N}$ such that the interior of the set*

$$
\mathcal{D}_N := \{(x_1, \ldots, x_N) \mid \forall \theta \in \mathbb{R}^d \colon \exists i = 1, \ldots, N \colon f(\theta, x_i) \neq 0\} \subseteq (\mathbb{R}^m)^N
$$

*is conull and dense in $(\mathbb{R}^m)^N$. If we moreover assume that $m = 1$, then $N$ can be chosen such that $\mathcal{D}_N$ contains all tuples $(x_1, \ldots, x_N)$ for which $x_1, \ldots, x_N$ are pairwise distinct.*

*Proof.* Fix $N \in \mathbb{N}$ such that $d + (m-1)N < mN$. Recall from [31, Chapter 4] that every definable set $X \subseteq \mathbb{R}^k$ for $k \in \mathbb{N}$ has a well defined *dimension* $\dim(X) \in \{-\infty, 0, 1, \ldots, k\}$ and that $\dim(X) = k$ if and only if $X$ has non-empty interior if and only if $X$ has non-zero Lebesgue measure [6, Theorem 2.5]. Note that the set

$$
X := \{(\theta, x) \in \mathbb{R}^d \times \mathbb{R}^m \mid f(\theta, x) = 0\} \subseteq \mathbb{R}^d \times \mathbb{R}^m
$$

as well as its slices

$$
X_\theta := \{x \in \mathbb{R}^m \mid f(\theta, x) = 0\} \subseteq \mathbb{R}^m
$$

for all $\theta \in \mathbb{R}^d$ are definable. Since each of the functions $f(\theta, -)$ for $\theta \in \mathbb{R}^d$ is analytic and not identically zero, the identity theorem implies that each $X_\theta$ for $\theta \in \mathbb{R}^d$ has empty interior. In particular, we have $\dim(X_\theta) \leq m - 1$ for each $\theta \in \mathbb{R}^d$. Note moreover that we have

$$
(\mathbb{R}^m)^N \setminus \mathcal{D}_N = \bigcup_{\theta \in \mathbb{R}^d} X_\theta^N \subseteq (\mathbb{R}^m)^N. \tag{11}
$$

The combination of Proposition 1.3 (iii), Proposition 1.5 and Corollary 1.6 (iii) in [31, Chapter 4] applied to (11) thus imply the inequality

$$
\dim((\mathbb{R}^m)^N \setminus \mathcal{D}_N) \leq d + (m-1)N < mN = \dim((\mathbb{R}^m)^N).
$$

Using Theorem 1.8 of [31, Chapter 4], we even get $\dim\left( \overline{(\mathbb{R}^m)^N \setminus \mathcal{D}_N} \right) = \dim((\mathbb{R}^m)^N \setminus \mathcal{D}_N) < \dim((\mathbb{R}^m)^N)$. In particular, $\overline{(\mathbb{R}^m)^N \setminus \mathcal{D}_N} \subseteq (\mathbb{R}^m)^N$ is a null set with empty interior. In other words: $\mathcal{D}_N^\circ \subseteq (\mathbb{R}^n)^N$ is conull and dense.

For the second part of the lemma, assume moreover that $m = 1$. Then each of the sets $X_\theta$ for $\theta \in \mathbb{R}^d$ is finite. The uniform finiteness theorem [15, Theorem 4.9] implies that the sets $(X_\theta)_{\theta \in \mathbb{R}^d}$ have uniformly finite cardinality, i.e. $\sup_{\theta \in \mathbb{R}^d} |X_\theta| < \infty$. The lemma then follows by picking $N = \sup_{\theta \in \mathbb{R}^d} |X_\theta| + 1$. $\square$

*Proof of Theorem 3.4.* By Lemma C.2 and Lemma C.3, for each $\theta \in \mathbb{R}^{d(\mathcal{A})}$, the analytic definable function

$$
\mathbb{R}^{d_0} \to \mathbb{R}, \quad x \mapsto \mathcal{N}_\theta(x) - f(x) \tag{12}
$$

is not identically zero. If $\mu$ is as in item (i) of Theorem 3.4, the theorem now follows from an application of Lemma C.4. Note that the identity theorem implies that (12) is non-zero outside of a null set. Since $\ell$ is a loss function, the function

$$
\mathbb{R}^{d_0} \to \mathbb{R}, \quad x \mapsto \ell(\mathcal{N}_\theta(x), f(x))
$$

is non-zero outside of a null set as well. This proves the theorem in the case that $\mu$ is as in [item (ii)](#) of Theorem 3.4. If $\mu$ is a convex combination of the two, the proof follows from the combination of the two arguments. $\qquad\square$

## D  Proof of Theorem 3.5

This section is devoted to the proof of Theorem 3.5 which follows classical proofs of the unviersal approximation theorem [16, 37, 44, 64]. For the particular case of $\tanh$ activations, the proof can be found in [21]. We do not claim originality and include the proof mainly for convenience of the reader. Before going into details, we outline the main ideas of the proof below.

**Idea of proof**

The proof starts from the special case of fitting linear polynomials in one variable with one hidden layer and then successively builds up generality (see Theorem D.4 and Figure 4). The starting point is that if $\psi$ is an analytic activation function and $x_0 \in \mathbb{R}$, then one can approximate a linear function with slope $\psi'(x_0)$ by neural networks with one hidden neuron using the formula $x \mapsto n\psi(x/n + x_0)$. Similarly, one can inductively approximate the $k$-th Taylor polynomial of $\psi$ using the formula $x \mapsto n^k\psi(x/n + x_0)$. By adding hidden neurons and compensating the lower order terms of the Taylor polynomial with previous approximations, one can thus approximate polynomials of degree $k$ using neural networks with one hidden layer and $k$ neurons. In order upgrade this to the multivariate case, we employ a linear algebra argument (see Lemma D.1) that allows us to express arbitrary multivariant polynomials as linear combinations of polynomials of the form $p(x_1, \ldots, x_n) = (a_1 x_1 + \cdots + a_n x_n)^k$. To extend our result to multiple hidden layers, we let the other layers approximate the identity and apply the Arzela-Ascoli theorem (see Lemma D.3) to achieve compatibility of the various approximations. Along the way we carefully keep track of the required number of hidden neurons.

**Lemma D.1.** *Fix integers $n \geq 1$ and $m \geq 0$. Denote by $\mathbb{R}[X_0, \ldots, X_m]_n \subseteq \mathbb{R}[X_0, \ldots, X_m]$ the real vector space of homogeneous polynomials of degree $n$ in $m + 1$ variables. Then, we have $\dim \mathbb{R}[X_0, \ldots, X_m]_n = \binom{n+m}{m}$ and moreover*

$$\mathbb{R}[X_0, \ldots, X_m]_n = \operatorname{span}\{(X_0 + a_1 X_1 + \cdots + a_m X_m)^n \mid a_1, \ldots, a_m \in \mathbb{R}\}.$$

To **avoid confusion about the dimensions**, we emphasize that this lemma consideres polynomials in $m + 1$ variables and not in $m$ variables.

*Proof of Lemma D.1.* We define the following index set.

$$\mathcal{S}(n, m) := \{\lambda = (\lambda_1, \ldots, \lambda_m) \in \mathbb{N}^m \mid 0 \leq \lambda_i \leq n, \lambda_1 + \cdots + \lambda_m \leq n\} \subseteq \{0, \ldots, n\}^m$$

For $\lambda \in \mathcal{S}(n, m)$, we furthermore write $\lambda_0 = n - \sum_{i=1}^m \lambda_i$. A basis of $\mathbb{R}[X_0, \ldots, X_m]_n$ is given by

$$\left(\binom{n}{\lambda_0, \ldots, \lambda_m} X_0^{\lambda_0} \cdots X_m^{\lambda_m}\right)_{\lambda \in \mathcal{S}(n,m)}, \tag{13}$$

where $\binom{n}{\lambda_0, \ldots, \lambda_m} = \frac{n!}{\lambda_0! \cdots \lambda_m!}$ denotes the multinomial coefficient. It is easy to see from a stars and bars argument that $\dim \mathbb{R}[X_0, \ldots, X_m]_n = |\mathcal{S}(n, m)| = \binom{n+m}{m}$.

To prove the second claim of the lemma, we define a linear map $P \colon \mathbb{R}[X_0, \ldots, X_m]_n \to \mathbb{R}[X_0, \ldots, X_m]_n$ on basis elements by the formula

$$P\left(\binom{n}{\lambda_0, \ldots, \lambda_m} X_0^{\lambda_0} \cdots X_m^{\lambda_m}\right) := (X_0 + \lambda_1 X_1 + \ldots + \lambda_m X^m)^n$$

$$= \sum_{\mu \in \mathcal{S}(n,m)} \binom{n}{\mu_0, \ldots, \mu_m} \lambda_1^{\mu_1} \cdots \lambda_m^{\mu_m} X_0^{\mu_0} \cdots X_m^{\mu_m}.$$

We introduce a relation $\prec$ on $\mathcal{S}(n, m)$ by defining that $\lambda \prec \mu$ if and only if $\lambda_r < \mu_r$ for some $r \in \{1, \ldots, m\}$ and $\lambda_i = \mu_i$ for all $i \in \{1, \ldots, m\} \setminus \{r\}$. In this case, we write $\Delta(\mu, \lambda) = \mu_r - \lambda_r$.

Using the basis (13) and the generalized Vandermonde determinant formula [13], we can calculate the determinant of $P$ as

$$\det(P) = \det\left( (\lambda_1^{\mu_1} \cdots \lambda_m^{\mu_m})_{\lambda,\mu \in \mathcal{S}(n,m)} \right) = \prod_{\substack{\lambda,\mu \in \mathcal{S}(n,m) \\ \lambda \prec \mu}} \Delta(\mu,\lambda) \neq 0.$$

In particular, $P$ is surjective and we have

$$\mathbb{R}[X_0,\ldots,X_m]_n \subseteq P(\mathbb{R}[X_0,\ldots,X_m]_n) \subseteq \operatorname{span}\{(X_0+a_1 X_1+\cdots+a_m X_m)^n \mid a_1,\ldots,a_m \in \mathbb{R}\}.$$

$\square$

**Definition D.2** (Uniform Norm). For a subset $X \subseteq \mathbb{R}^n$ and a function $f\colon \mathbb{R}^n \to \mathbb{R}$, we write

$$\|f\|_X := \sup_{x \in X} |f(x)|.$$

For a sequence $(f_j)_{j \in \mathbb{N}}$ of functions $\mathbb{R}^n \to \mathbb{R}$, we say that $(f_j)_{j \in \mathbb{N}}$ *converges to $f$ uniformly on compact sets*, if for every compact set $K \subseteq \mathbb{R}^n$, we have

$$\|f_j - f\|_K \xrightarrow{j \to \infty} 0.$$

For approximating the identity with redundant hidden layers, we will also need the following elementary application of the Arzelà–Ascoli theorem.

**Lemma D.3.** *Let $n, m, k \in \mathbb{N}$ and let $f_j, f_\infty \colon \mathbb{R}^m \to \mathbb{R}^k$ and $g_j, g_\infty \colon \mathbb{R}^n \to \mathbb{R}^m$ for all $j \in \mathbb{N}$ be continuous functions such that $f_j \xrightarrow{j \to \infty} f_\infty$ and $g_j \xrightarrow{j \to \infty} g_\infty$ uniformly on compact sets. Then we have $f_j \circ g_j \xrightarrow{j \to \infty} f_\infty \circ g_\infty$ uniformly on compact sets.*

*Proof.* Fix a compact subset $K \subseteq \mathbb{R}^n$ and $\varepsilon > 0$. By the Arzelà–Ascoli theorem, there is an $R > 0$ satisfying

$$\bigcup_{j \in \mathbb{N} \cup \{\infty\}} g_j(K) \subseteq B_R(0).$$

Again by the Arzelà–Ascoli theorem, there is $\delta > 0$ such that for all $j \in \mathbb{N} \cup \{\infty\}$ and $x, y \in B_R(0)$ with $|x - y| < \delta$, we have $|f_j(x) - f_j(y)| < \frac{\varepsilon}{2}$. Now choose a large enough $N \in \mathbb{N}$ such that for all $j \in \mathbb{N}$ with $j \geq N$, we have

$$\|f_j - f_\infty\|_{B_R(0)} < \frac{\varepsilon}{2}$$

and

$$\|g_j - g_\infty\|_K < \delta.$$

Then we have

$$|f_j(g_j(x)) - f_\infty(g_\infty(x))| \leq |f_j(g_j(x)) - f_j(g_\infty(x))| + |f_j(g_\infty(x)) - f_\infty(g_\infty(x))| < \varepsilon$$

for all $x \in K$ and $j \in \mathbb{N}$ with $j \geq N$. $\square$

**Theorem D.4.** *Let $n \in \mathbb{N}$ and $\psi \colon \mathbb{R} \to \mathbb{R}$ be a $C^n$-function with $\psi^{(n)} \not\equiv 0$ and let $p \colon \mathbb{R}^m \to \mathbb{R}^r$ be a polynomial of degree at most $n$. Let $\mathcal{A} = (d_0,\ldots,d_k,\psi)$ be a neural network architecture with $d_0 = m$ input neurons, $d_k = r$ output neurons and activation function $\psi$. Suppose that $\mathcal{A}$ is sufficiently big in the sense that*

*(i) there is a hidden layer $0 < i < k$ of size $d_i \geq r\left(\binom{n+m}{m} - m\right)$,*

*(ii) all previous hidden layers have size $\min(d_0,\ldots,d_{i-1}) \geq d_0$,*

*(iii) all subsequent hidden layers have size $\min(d_{i+1},\ldots,d_k) \geq d_k$.*

*Then $p$ can be approximated uniformly on compact sets by realization functions $\mathcal{N}_\theta^{\mathcal{A}} \colon \mathbb{R}^m \to \mathbb{R}^r$ for $\theta \in \mathbb{R}^d$.*

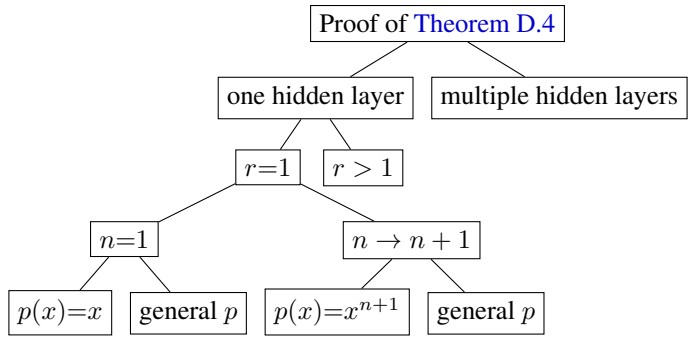

Figure 4: Structure of the special cases in the proof of Theorem D.4.

*Proof.* We prove the theorem by considering several successively more general cases. The structure of the proof is visualized in Figure 4.

Throughout the proof, we denote for $j \in \mathbb{N}$ by

$$\mathfrak{R}_{m,j,r}^{\psi} \coloneqq \{\mathcal{N}_{\theta}^{(m,j,r,\psi)} \mid \theta \in \mathbb{R}^{(m+1)j+(j+1)r}\} \subseteq C^n(\mathbb{R}^m, \mathbb{R}^r)$$

the set of all realization functions of shallow neural networks with $m$ input neurons, $j$ hidden neurons, $r$ output neurons and activation $\psi$. Note that

$$\mathfrak{R}_{m,i,r}^{\psi} + \mathfrak{R}_{m,j,r}^{\psi} = \mathfrak{R}_{m,i+j,r}^{\psi} \tag{14}$$

for all $i, j \in \mathbb{N}$. We moreover denote by $\overline{\mathfrak{R}}_{m,j,r}^{\psi} \subseteq C(\mathbb{R}^m, \mathbb{R}^r)$ the set of all functions $f \colon \mathbb{R}^m \to \mathbb{R}^r$, which can be approximated uniformly on compact subsets by elements of $\mathfrak{R}_{m,j,r}^{\psi}$. For $a, x \in \mathbb{R}^m$, we denote by $\langle x, y \rangle = \sum_{i=1}^{m} a_i x_i$ their inner product.

**Case 1:** We have $r = 1$ and $\mathcal{A} = (m, \binom{n+m}{m} - m, 1, \psi)$.

We prove Case 1 by induction on $n$.

We start with the case $n = 1$. We first consider the case that $p$ is equal to the identity function $\mathrm{id}_{\mathbb{R}} \colon \mathbb{R} \to \mathbb{R}, \quad \mathrm{id}_{\mathbb{R}}(x) = x$. By the assumptions on $\psi$, there exists $x_0 \in \mathbb{R}$ with $\alpha \coloneqq \psi'(x_0) \neq 0$. Observe that the functions $(f_j)_{j \in \mathbb{N}} \subseteq \mathfrak{R}_{1,1,1}^{\psi}$ defined by

$$f_j(x) = \frac{j}{\alpha}(\psi(x/j + x_0) - \psi(x_0)), \quad x \in \mathbb{R}$$

converge along $j \to \infty$ to $\mathrm{id}_{\mathbb{R}}$ uniformly on every compact subset $I \subseteq \mathbb{R}$. Indeed, using continuity of $\psi'$, we have for every $j \in \mathbb{N}$ that

$$
\begin{aligned}
\sup_{x \in I} |f_j(x) - x| &= \sup_{x \in I} \left| \frac{j}{\alpha} \int_{x_0}^{x_0 + \frac{x}{j}} \psi'(t) dt - x \right| \\
&= \sup_{x \in I} \left| \int_{x_0}^{x_0 + \frac{x}{j}} \left( \frac{j}{\alpha} \psi'(t) - j \right) dt \right| \\
&\leq \frac{j}{|\alpha|} \sup_{x \in I} \int_{x_0}^{x_0 + \frac{x}{j}} |\psi'(t) - \alpha| \, dt \\
&\leq \frac{1}{|\alpha|} \underbrace{\sup_{x \in I} |x|}_{< \infty} \underbrace{\sup_{t \in [x_0, x_0 + \frac{x}{j}]} |\psi'(t) - \alpha|}_{\to 0} \\
&\xrightarrow{j \to \infty} 0.
\end{aligned}
$$

This proves that $\mathrm{id}_{\mathbb{R}} \in \overline{\mathfrak{R}}_{1,1,1}^{\psi}$.

Now let
$$p\colon \mathbb{R}^m \to \mathbb{R}, \quad p(x) = \langle a, x \rangle + a_0 = a_1 x_1 + \cdots + a_m x_m + a_0$$
be an arbitrary polynomial of degree at most one with $a \in \mathbb{R}^m$ and $a_0 \in \mathbb{R}$. Choose a sequence $(f_j)_{j \in \mathbb{N}} \subseteq \mathfrak{R}^\psi_{1,1,1}$ converging to $\mathrm{id}_\mathbb{R}$ uniformly on compact subsets. Since $p$ is continuous and $f_j \xrightarrow{j \to \infty} \mathrm{id}_\mathbb{R}$ uniformly on compact sets, the sequence $(\tilde{f}_j)_{j \in \mathbb{N}} \subseteq \mathfrak{R}^\psi_{m,1,1}$ defined by

$$\tilde{f}_j(x) := f_j(\langle a, x \rangle + a_0)$$

converges to $p$ uniformly on compact sets. This proves that $p \in \overline{\mathfrak{R}}^\psi_{m,1,1}$. Noting that $\binom{1+m}{m} - m = 1$, this concludes the case $n = 1$.

We now prove Case 1 for $n + 1$, assuming that we have proved Case 1 for $n$. We first consider the case that $p$ is given by the $(n+1)$-th power function $\mathrm{id}_\mathbb{R}^{n+1} \colon \mathbb{R} \to \mathbb{R}, \quad \mathrm{id}_\mathbb{R}^{n+1}(x) = x^{n+1}$. By the assumptions on $\psi$, there exists $x_0 \in \mathbb{R}$ with

$$\alpha := \frac{\psi^{(n+1)}(x_0)}{(n+1)!} \neq 0. \tag{15}$$

We denote by

$$T_n \psi \colon \mathbb{R} \to \mathbb{R}, \quad T_n \psi(x) := \sum_{j=0}^{n} \frac{\psi^{(j)}(x_0)}{j!}(x - x_0)^j \tag{16}$$

the Taylor approximation of $\psi$ at $x_0$ of order $n$. We define functions $(g_j)_{j \in \mathbb{N}} \subseteq C^{n+1}(\mathbb{R}, \mathbb{R})$ by

$$g_j(x) = \frac{j^{n+1}}{\alpha}(\psi(x_0 + x/j) - T_n \psi(x_0 + x/j)), \quad x \in \mathbb{R} \tag{17}$$

and claim that

$$\sup_{x \in I} |g_j(x) - x^{n+1}| \xrightarrow{j \to \infty} 0 \tag{18}$$

for every compact subset $I \subseteq \mathbb{R}$.

Indeed, by the Lagrange form of the remainder in Taylor's formula, for every $j \in \mathbb{N}$ there is a $\xi_j \in [x_0, x_0 + x/j]$ satisfying

$$\sup_{x \in I} |g_j(x) - x^{n+1}| = \sup_{x \in I} \left| \frac{j^{n+1}}{\alpha} \frac{\psi^{(n+1)}(\xi_j)}{(n+1)!} \left(\frac{x}{j}\right)^{n+1} - x^{n+1} \right|$$

$$= \sup_{x \in I} \left| \left(\frac{\psi^{(n+1)}(\xi_j)}{\psi^{(n+1)}(x_0)} - 1\right) x^{n+1} \right|$$

$$\xrightarrow{j \to \infty} 0,$$

where at the last step we have used continuity of $\psi^{(n+1)}$ and compactness of $I$.

Now fix an arbitrary polynomial $p\colon \mathbb{R}^m \to \mathbb{R}$ of degree at most $n + 1$. We write $p = p_0 + p_1$ where $p_0$ is a homogeneous polynomial of degree $n + 1$ and $p_1$ is a polynomial of degree at most $n$. By Lemma D.1 (applied to $n \leftarrow n + 1$ and $m \leftarrow m - 1$), we can find $a^{(i)} \in \mathbb{R}^m$ for $i = 1, \ldots, \binom{n+m}{m-1}$ such that

$$p_0(x) = \sum_{i=1}^{\binom{n+m}{m-1}} (\langle a^{(i)}, x \rangle)^{n+1}, \quad \forall x \in \mathbb{R}^m. \tag{19}$$

Fix an increasing sequence $(K_j)_{j \in \mathbb{N}}$ of compact subsets of $\mathbb{R}^m$ such that $\mathbb{R}^m = \bigcup_{j \in \mathbb{N}} K_j$. Since $\psi$ satisfies the assumptions of the theorem for $n + 1$, then it does for $n$. In particular, for every $j \in \mathbb{N}$, we can find a function $q_j \in \mathfrak{R}^\psi_{m, \binom{n+m}{m} - m, 1}$ such that

$$\sup_{x \in K_j} \left\| q_j(x) - p_1(x) + \sum_{i=1}^{\binom{n+m}{m-1}} \frac{j^{n+1}}{\alpha} T_n \psi(\langle a^{(i)}, x \rangle) \right\| < \frac{1}{j}, \tag{20}$$

where $T^n \psi$ is defined in (16).

We now define a sequence of functions $(f_j)_{j \in \mathbb{N}} \subseteq C^{n+1}(\mathbb{R}^m, \mathbb{R})$ by

$$f_j(x) = q_j(x) + \sum_{i=1}^{\binom{n+m}{m-1}} \frac{j^{n+1}}{\alpha} \psi \left( x_0 + \frac{1}{j}(\langle a^{(i)}, x \rangle) \right),$$

where $\alpha, x_0 \in \mathbb{R}$ are defined as in (15).

Note that by definition, the first summand of $f_j$ defines an element of $\mathfrak{R}^\psi_{m, \binom{n+m}{m}-m, 1}$ while the second summand defines an element of $\mathfrak{R}^\psi_{m, \binom{n+m}{m-1}, 1}$. Thus, by (14), $f_j$ defines an element of $\mathfrak{R}^\psi_{m, \binom{n+1+m}{m}-m, 1}$.

Now let $K \subseteq \mathbb{R}^m$ be a compact subset and $j \in \mathbb{N}$. Using (17) and (19) at the second step, and the combination of (18) and (20) at the last step, we get

$$\sup_{x \in K} |f_j(x) - p(x)|$$

$$= \sup_{x \in K} \left| q_j(x) + \sum_{i=1}^{\binom{n+m}{m-1}} \frac{j^{n+1}}{\alpha} \psi \left( x_0 + \frac{1}{j}(\langle a^{(i)}, x \rangle) \right) - p_0(x) - p_1(x) \right|$$

$$= \sup_{x \in K} \left| q_j(x) - p_1(x) + \sum_{i=1}^{\binom{n+m}{m-1}} \frac{j^{n+1}}{\alpha} T^n \psi(\langle a^{(i)}, x \rangle) + \sum_{i=1}^{\binom{n+m}{m-1}} \left( g_j(\langle a^{(i)}, x \rangle) - (\langle a^{(i)}, x \rangle)^{n+1} \right) \right|$$

$$\leq \sup_{x \in K} \left| q_j(x) - p_1(x) + \sum_{i=1}^{\binom{n+m}{m-1}} \frac{j^{n+1}}{\alpha} T^n \psi(\langle a^{(i)}, x \rangle) \right| + \sup_{x \in K} \sum_{i=1}^{\binom{n+m}{m-1}} \left| g_j(\langle a^{(i)}, x \rangle) - (\langle a^{(i)}, x \rangle)^{n+1} \right|$$

$$\xrightarrow{j \to \infty} 0.$$

This proves that $p \in \overline{\mathfrak{R}}^\psi_{m, \binom{n+1-m}{m}-m, 1}$ and finishes the proof of Case 1.

**Case 2:** We have $\mathcal{A} = (m, r \left( \binom{n+m}{m} - m \right), r, \psi)$. This case follows directly from Case 1 by applying Case 1 to each of the components $p_1, \ldots, p_r$ of $p = (p_1, \ldots, p_r) \colon \mathbb{R}^m \to \mathbb{R}^r$ and stacking the neural network architectures used in the approximation on top of each other.

**Case 3:** We have a general architecture $\mathcal{A} = (m, d_1, \ldots, d_{k-1}, r, \psi)$ as in the statement of the theorem.

We fix $0 < i < k$ such that

(i) $d_i \geq r \left( \binom{n+m}{m} - m \right)$,

(ii) $\min(d_0, \ldots, d_{i-1}) \geq d_0$,

(iii) $\min(d_{i+1}, \ldots, d_k) \geq d_k$.

By Case 2, there is a sequence $(f_j^{(i)})_{j \in \mathbb{N}} \subseteq \mathfrak{R}_{m, r(\binom{n+m}{m}-m), r}$ converging to $p$ uniformly on compact subsets of $\mathbb{R}^m$. By a componentwise application of Case 1, for any $0 \leq l < i$, there is a sequence $(f_j^{(l)})_{j \in \mathbb{N}} \subseteq \mathfrak{R}^\psi_{m, m, m}$ converging to $\mathrm{id}_{\mathbb{R}^m}$ uniformly on compact sets. Similarly, for any $i < l \leq k$, there is a sequence $(f_j^{(l)})_{j \in \mathbb{N}} \subseteq \mathfrak{R}^\psi_{r, r, r}$ converging to $\mathrm{id}_{\mathbb{R}^r}$ uniformly on compact sets. By Lemma D.3, we have

$$f_j^{(k)} \circ \cdots \circ f_j^{(0)} \xrightarrow{j \to \infty} p$$

uniformly on compact subsets of $\mathbb{R}^m$. Moreover, by the assumptions on $d_0, \ldots, d_k$ we have

$$f_j^{(k)} \circ \cdots \circ f_j^{(0)} \in \mathfrak{R}^\psi_{r, r, r} \circ \cdots \circ \mathfrak{R}^\psi_{m, r(\binom{n+m}{m}-m), r} \circ \cdots \circ \mathfrak{R}^\psi_{m, m, m} \subseteq \{ \mathcal{N}^\mathcal{A}_\theta \mid \theta \in \mathbb{R}^{d(\mathcal{A})} \},$$

see Figure 5 for an illustration. This finishes the proof of the theorem. $\qquad\square$

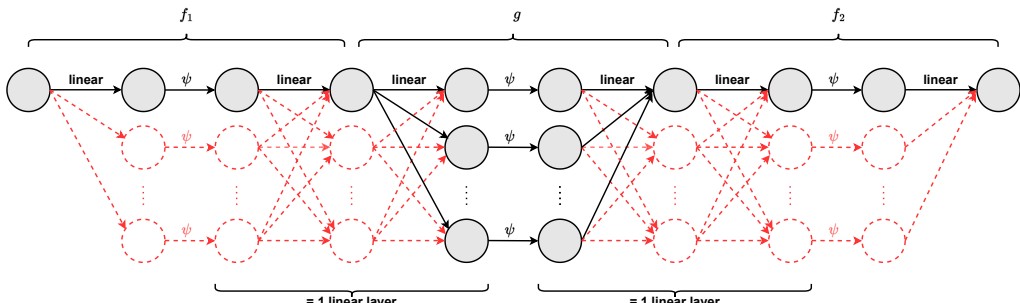

Figure 5: An illustration how a neural network with 3 hidden layers can realize functions of the form $f_2 \circ g \circ f_1$ where $g$ is the realization of a shallow neural network with $n$ neurons and $f_i$ are realizations of shallow neural networks with on hidden neuron. Dotted lines correspond to weights that are set to zero.

*Proof of Theorem 3.5.* Theorem 3.5 follows immediately from Theorem D.4 and the fact that $\mu$ is a probability measure since uniform convergence on compact sets implies $L^1$-convergence with respect to probability measures. $\qquad\square$

**Remark D.5.** The bounds on the required number of neurons in Theorem 3.5 are not sharp. In many cases, one can approximate polynomials with far fewer neurons. This can be done by following the steps in the proof of Theorem D.4 in detail and depends on several factors such as

   (i) the number of nonzero homogeneous parts of the polynomial,

  (ii) how many nonzero coefficients are needed to express the homogenous polynomials in the basis from Lemma C.4,

 (iii) the zeros of the Taylor coefficients of the activation function.

Moreover, the number of neurons can be reduced drastically if one allows for deep neural networks and rewrites the polynomial as a composition of easier polynomials. The polynomial $p(x,y) = x^7 y^9$ for example can be rewritten as the composition of the functions $(x,y) \mapsto (x^7, y^9)$ and $(x,y) \mapsto xy$ and can therefore be approximated by realization functions of the architecture $\mathcal{A} = (2, 16, 4, 1, \psi)$ for any SAD activation function $\psi$.

# E   Experimental Details

In this section, we detail the experimental setup. All experiments were conducted on a single Tesla V100 GPU, with each training run requiring between 10 and 20 minutes. Overall, the computational demands of our study remain modest.

**Polynomial target function**

We train our neural networks to approxomate the polynomial target function $p \colon [a,b]^{d_0} \to \mathbb{R}$ where $[a,b]^{d_0} = [-1,1]^{d_0}$. The target functions differ based on the input dimension $d_0$:

   (i) for $d_0 = 1$,
$$p(x_1) = x_1^{10} - 2x_1^8 + 2x_1^5 + 3x_1^3 - 2x_1^2 + 5,$$

  (ii) for $d_0 = 2$,
$$p(x_1, x_2) = x_2^5 - x_1^3 x_2^2 - 4x_1^2 x_2 + 3x_1^3 - x_2^2 + x_1 + 2,$$

and

(iii) for $d_0 = 4$,

$$p(x_1, x_2, x_3, x_4) = x_1^6 x_4^5 + x_2^6 - x_1^3 x_2^2 x_3 + x_4^2 - 4x_3^4 x_2^4 + 3x_4^3 x_2^3 - x_3^2 x_1 + x_3 + 3.$$

In the one-dimensional case, the neural network architecture consists of three hidden layers with widths $(d_1, d_2, d_3) = (10, 20, 10)$. For the higher-dimensional settings we increase the dimensions to $(d_1, d_2, d_3) = (20, 40, 20)$.[4] For gradient descent, we use the full training dataset of 10,000 randomly sampled points at each step. For Adam, batches of 100 samples are drawn from the same dataset. We use a learning rate of 0.001 for gradient descent and 0.005 for Adam. Note that the plotted loss corresponds to the training loss. Our theoretical results, indeed, show that the norm of the neural network parameters diverges while the training loss converges to zero. To reduce noise, we plot the exponential moving average of the training loss with smoothing factor $\alpha = 0.95$, averaged over 20 independent random initializations. We also construct a test set as a uniform grid over the domain. The test loss exhibits a similar qualitative behavior. We include the plots of the test losses in the GitHub repository https://github.com/deeplearningmethods/sad.

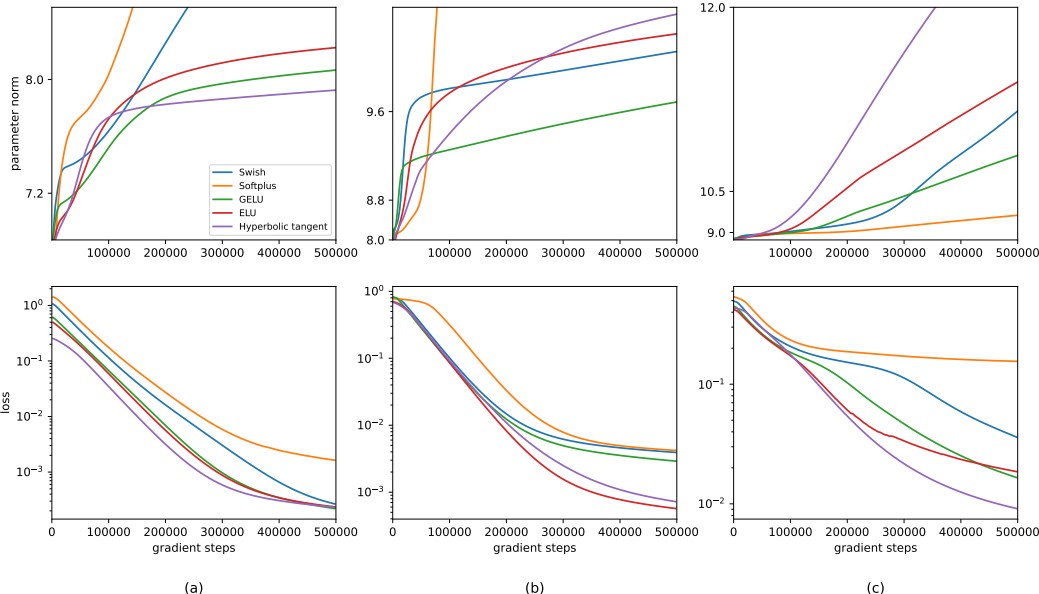

Figure 6: Approximation of polynomial target functions using different activation functions and GD algorithm. From left to right: 1-dimensional input case, 2-dimensional input case, 4-dimensional input case. The $y$-axis is rescaled exponentially in order to illustrate the logarithmic growth of the parameter norms.

**Real world learning task**

To solve the PDE problems we use the Deep Kolmogorov method. This approach approximates the solution using a neural network, leveraging the Feynman–Kac representation of the PDE solution. Specifically, the neural network is trained to match the expected terminal value of the underlying stochastic differential equation (SDE), estimated via Monte Carlo sampling. The method provides a generalizable solution across input space, enabling instant evaluations of the PDE once the network is trained. This is particularly advantageous in high-dimensional settings, where traditional solvers suffer the curse of dimensionality. We refer to [5] for a comprehensive description of the method.

In the case of the Heat PDE we consider a neural network with architecture $(d_0, d_1, d_2, d_3, d_4) = (10, 50, 50, 50, 1)$ while for the Black-Scholes PDE $(d_0, d_1, d_2, d_3, d_4) = (10, 200, 300, 200, 1)$. For the latter, we set the interest rate $r = 0.05$, the cost

---

[4]The reader might notice that the architecture in the four dimensional input case does not satisfy the crude bounds of (i) from Theorem 3.5. However, as pointed out in Remark D.5, these are only worst case bounds. In specific examples like these, one can approximate the given polynomials with significantly smaller architectures.

of carry $c = 0.01$, the strike price $K = 100$ and define the volatility vector as $\sigma = (0.1000, 0.1444, 0.1889, 0.2333, 0.2778, 0.3222, 0.3667, 0.4111, 0.4556, 0.5000)$. The loss plots in Figure 3 report the relative mean squared error between the neural network prediction and the reference solutions. For the Heat PDE, the solution is available in closed form, while for the Black–Scholes PDE we use a high-precision Monte Carlo approximation by averaging over 1000 independents rounds, each consisting of 1024 sample paths. Importantly, the reference solutions are not used during training. At each training step, a single Monte Carlo sample is employed as the target solution. We train the neural networks using the Adam optimizer with a learning rate of 0.005.

For the MNIST classification task, we use a neural network with architecture $(d_0, d_1, d_2, d_3) = (784, 256, 256, 10)$ trained with a standard cross-entropy loss. Optimization is performed with the Adam optimizer and a learning rate of 0.001. As in the previous experiments, we report the training loss in the figure. Training is stopped after 30000 steps to prevent overfitting, resulting in a final test accuracy of $97.93\%$.

In both the MNIST and PDE examples, we smooth the plots by plotting an exponential moving average of the loss with smoothing factor $\alpha = 0.95$, averaged over 5 independent random initializations.

