# OpenReview forum: "SAD Neural Networks: Divergent Gradient Flows and Asymptotic Optimality via o-minimal Structures"
_NeurIPS.cc/2025/Conference — NeurIPS 2025 poster_

### Official Review · Reviewer_hwZX · 2025-06-16

**Clarity:** 4
**Significance:** 2
**Originality:** 2
**Rating:** 5
**Confidence:** 4

**Summary:**

In Section 2, the authors recall definitions of o-minimal structures (families of subsets satisfying certain algebraic properties) and functions that are definable with respect to such structures, as well as identify many examples of activation functions which are definable. The authors conclude the section by stating their dichotomy theorem (Theorem 2.8): a network with definable activations, optimized using gradient flow to fit a definable target function using a definable loss on either an empirical or absolutely continuous input measure, must either converge to a critical point of the loss or diverge with loss approaching an asymptotic critical value.

In Section 3, the authors consider a special case of definable functions, namely sublinear analytic definable (SAD), which are exemplified by typical activations and losses. They state their Theorem 3.4, which says that there are no global minima for the problem of fitting a network with SAD activations to a polynomial of degree at least 2 on a sufficiently large dataset. Theorem 3.4 is counterbalanced by Theorem 3.5, which says that any polynomial can be approximated arbitrarily well by a sufficiently large (with respect to the degree of the polynomial) neural network on compact sets. The results are synthesised into the main result of the paper, Corollary 3.6, which states that in fitting a sufficiently large neural network to a polynomial of degree 2 using gradient flow on a sufficiently large dataset starting from a point with sufficiently small loss, the parameters necessarily diverge.

In Section 4, the authors provide some numerical support for their divergence theorem.

**Questions:**

I am willing to raise my score to an accept if the authors can answer the following questions.
1. Why is the result significant, and how is it a significant advance over prior work in this area?
2. In a couple of paragraphs, can you summarise how the assumptions made imply the result?

**Ethical Concerns:**

["NO or VERY MINOR ethics concerns only"]

**Final Justification:**

I was already very impressed by this paper. My only concern was the significance of the results, however these have been adequately addressed by the authors in their rebuttal.

**Limitations:**

Yes

**Quality:**

4

**Strengths And Weaknesses:**

Strengths:
 - The paper is exceptionally well-written and easy to read.
 - The approach taken in this paper of using abstract, formal properties to prove theorems is underutilised in the literature. As a consequence, the theorems are relatively general in their assumptions compared to most other papers on the optimisation of deep networks.

Weaknesses:
 - The paper lacks any "idea of proof". It is not clear at all in the paper how the assumptions made are used in the derivation of the theorems. The proofs in the appendix, while complete as far as I can tell, also seem to lack a high level idea of the proof.
 - There is not much specific said about the relation of this work to prior works. In particular, as someone with little knowledge about the employment of o-minimality in optimisation, it is difficult for me to judge the novelty of the work.
 - It is not made clear by the authors why the result is significant.

---

> ### Author Rebuttal · Authors · 2025-07-30
>
> We thank the reviewer for the constructive comments and for the opportunity to clarify the significance and methodology of our work.
>
> Regarding the significance of our results: The core contribution of this work is to introduce new mathematical tools — specifically, o-minimal geometry and definable dimension theory — into the analysis of divergence phenomena in neural network optimization. These tools allow us to study optimization dynamics in a general setting with minimal reliance on the explicit functional form of the neural network.\
> The principal novelty lies in Theorem 3.4 and the resulting Corollary 3.6, which together provide, to our knowledge, the first proof of divergence of parameters during training of neural networks using o-minimal techniques. While divergence phenomena are empirically observed and have been studied in special cases (e.g., shallow networks with specific activations), our results generalize and formalize this behavior in a broad setting that includes many commonly used activation functions. In particular:
>    1. Theorem 3.4 establishes the non-existence of global minimizers for networks with SAD activations trained on polynomials of degree $\geq 2$ over large datasets. Only very special cases of this result were previously known, e.g., [28] for shallow networks fitting $x^2$, and [De Ryck, Tim, Samuel Lanthaler, and Siddhartha Mishra. On the approximation of functions by tanh neural networks. Neural Networks 143:732-750, 2021] for tanh shallow networks.
>    2. Corollary 3.6 then shows that parameter divergence is inevitable in such settings, providing rigorous theoretical support for an empirical phenomenon that had remained poorly understood in generality.
>
> Additionally, we isolate a flexible and practically relevant class of activation functions — SAD (Sublinear Analytic Definable) — that both allows unified analysis and includes most smooth activations used in practice.
> Our findings suggest an important insight: parameter divergence is not necessarily harmful. On the contrary, it may be required to achieve low loss in certain regimes.
>
> A significant advance over prior work is our abstract, theory-driven approach. While most previous analyses are limited to specific architectures, losses, or activation functions and rely on direct computation, our use of o-minimality allows us to obtain general results in a uniform, mathematically principled way. This opens the door to future applications of definable geometry in deep learning theory and contributes to building a more robust understanding of neural network optimization.\
> In the revision, we will expand our discussion of related work, including references to results known to experts (e.g., the convergence part of Theorem 2.8) and clarify which results are new or more general than what has appeared before. More information about other results in the literature can be found in the rebuttals to other reviewers.
>
> We now answer the second question by giving a summary of the ideas of proof.
> 1. **Proof of Theorem 2.8**: The proof of Theorem 2.8 to a large extend follows standard techniques and is only given for convenience of the reader. We will point out in the revision that this theorem was probably well-known to the community and that our main contributions are Theorems 3.4 and 3.5 and our resulting main result Corollary 3.6. Having said this, the proof idea for Theorem 2.8 is the following: Existence and uniqueness of the gradient flow for $\mathcal L$ follow from the fact that $\mathcal L$ is $C^1$ and has locally Lipschitz gradients. The latter assumption is checked in Appendix A. Now one can distinguish two cases: Either the gradient flow visits some bounded set infinitely often or it does not. In the first case, we use Kurdyka's definable Lojasiewicz inequality and follow a standard argument to prove convergence to a critical point (see Lemma B.1). In the second case, the parameters diverge by assumption. We check that the limit is a generalized critical value using the fact that $\mathcal L$ is bounded below and therefore has square-integrable gradients along any gradient flow trajectory (see Lemma B.2).
> 2. **Proof of Theorem 3.4**: The proof of Theorem 3.4 is the main part where o-minimal structures are used in a new way which cannot be found in the neural network literature. The starting point is that the sublinearity condition in our SAD activation functions also implies that the neural network responses $\mathcal N_\theta^\mathcal A$ are sublinear (Lemma C.2). In particular, it cannot be equal as a function to a polynomial $f$ of degree at least $2$ (Lemma C.3). This does not explain why $\mathcal L$ does not have any zeros yet since $\mathcal N_\theta^\mathcal A$ and $f$ could be equal on the dataset or data distribution for some $\theta$. The main bulk of the proof now exploits the remarkable rigidity properties of definable analytic functions to show that this cannot happen (Lemma C.4). We treat the one-dimensional and multi-dimensional input cases separately. In the one-dimensional case, the combination of the identity theorem for analytic functions and the uniform finiteness theorem from o-minimal geometry imply that for any $\theta$, $\mathcal N_\theta^\mathcal A$ and $f$ cannot be equal on a sufficiently large finite dataset where the required size can be made independent of $\theta$. In the multidimensional case, the conclusion can fail for very degenerate datasets (for example if all datapoints lie on a line along which the polynomial is linear). But by combining the ideas from the one-dimensional case with dimension theory for definable sets, we show that the space of such "bad datasets" has strictly smaller dimension than the space of all datasets and therefore contains an open dense subset of full Lebesgue measure.
> 3. **Proof of Theorem 3.5**: The proof of Theorem 3.5 starts from the special case of fitting linear polynomials in one variable with one hidden layer and then successively builds up generality (see Theorem D.4). The starting point is that if $\psi$ is the activation function and $x_0\in \mathbb{R}$, then one can approximate a linear function with slope $\psi'(x_0)$ by neural networks with one hidden neuron using the formula $x \mapsto n \psi(x/n +x_0)$. Similarly, one can inductively approximate the $k$-th Taylor polynomial of $\psi$ using the formula $x\mapsto n^k \psi(x/n+x_0)$. By adding hidden neurons and compensating the lower order terms of the Taylor polynomial with previous approximations, one can thus approximate polynomials of degree $k$ using neural networks with one hidden layer and $k$ neurons. To upgrade this to the multivariate case, we employ a linear algebra argument (see Lemma D.1) that allows us to express arbitrary multivariate polynomials as linear combinations of polynomials of the form $p(x_1,\dotsc,x_n)=(a_1 x_1 + \dotsb + a_n x_n)^k$. To extend our result to multiple hidden layers, we let the other layers approximate the identity and apply the Arzela-Ascoli theorem (Lemma D.3) to achieve compatibility of the various approximations. Along the way we carefully keep track of the required number of hidden neurons.
> 4. **Proof of Corollary 3.6**: Corollary 3.6 is an immediate consequence of the previous results.

---

> > ### Comment · Reviewer_hwZX · 2025-08-01
> >
> > Thank you for your rebuttal, I will raise my score to 5, Accept.

---

### Official Review · Reviewer_7NML · 2025-06-26

**Clarity:** 4
**Significance:** 4
**Originality:** 3
**Rating:** 5
**Confidence:** 5

**Summary:**

The paper discusses gradient flows (GF) for neural networks with definable activations and loss functions. First, using Kurdyka-Lojasiewicz inequality, the author gives a dichotomy of convergence of GF for such neural networks. This is basically a summary of existing results. Second, the author studies  fitting polynomials (usually with degree $\ge 2$) with SAD networks. He/she shows that for sufficiently large model and sufficiently large sample size, we have 1) 0 is an asymptotic critical value of the loss, 2) 0 is not a critical value of the loss, and thus 3) GF diverges to infinity. Experiments are presented and limitations of the works are discussed.

**Questions:**

1. Is it Okay to mention someone’s real name in the paper??
2. Even if GD is not used widely in practice, I suggest performing experiments using GD. In fact, this is necessary based on the authors’ plan for the paper, as experiments “are intended as a proof of concept”, mentioned in 1.3.
3. Some typos:
    Typo at line 188-189: We write $d\mu(x) = p(x) dx$ if $\mu$ is given by $\mu(A) = \int_A p(x) dx$
    Typo at line 697: should mention that the $p_0, p_1, …$, etc. are homogeneous polynomials.
    Typo at line 827: should be $T_n$ instead of $T^n$ in equation (20).

**Ethical Concerns:**

["NO or VERY MINOR ethics concerns only"]

**Final Justification:**

Final score: 5. The paper is excellent overall and is clearly written.

**Limitations:**

Yes. Discussed in the article.

**Quality:**

3

**Strengths And Weaknesses:**

Strengths:
1.  The paper has very clear structure and provides sufficient details for readers.
2.  The paper introduces a useful tool, i.e., theory in o-minimal structure, to the study of machine learning.   I believe that this tool greatly helps us understand the behaviors of neural networks.
3.  The results shown in the paper, especially the divergence of GF for fitting polynomials, captures, and explains the phenomenon in application in an unambiguous way, meanwhile having sufficient generality in mathematics.

Weakness:
1.  My major concern is that the experiments are performed using SGD and Adam, which usually differ a lot from GF/GD.
2.  My second concern is about contribution. The authors mainly discuss the dichotomy of GF (see 1.1.Contribution) in analytic setting. But when the loss is analytic, this behavior of GF has been proven by previous works, such as [1]. And even for general definable functions, it follows quickly from KL inequality. So it might not be appropriate to put it as a contribution of this paper.
3. I found some typos. See below.

[1] P. Absil, R. Mahony, B. Andrew, “Convergence of the Iterates of Descent Methods for Analytic Cost Functions”, SIAM Journal on Optimization, 2005, 16(2), pp. 531-547.

---

> ### Author Rebuttal · Authors · 2025-07-30
>
> We thank the reviewer for the very thorough reading and the positive and constructive report. Below, we address the concerns and questions raised in detail.
>
> - Concerning the first raised weakness: We agree that SGD and Adam differ from GF and GD in general.  As noted by the reviewer, the main contribution of this work is the development of new mathematical tools for the analysis of training dynamics of neural networks. While we do believe that our results indicate the practical insight that diverging parameters can sometimes be desirable for performance, we acknowledge that our experiments are rather limited and mainly serve as a proof of concept. To motivate the use of Adam in the experiments we point out that the square-integrability of the gradient norms (see equation (7)) implies that the gradient norms decay like $O(1/\sqrt{t})$ and thus the model parameters can grow at most like $O(\sqrt{t})$. Adam’s adaptive rescaling effectively normalizes gradients to $O(1)$ per coordinate, making the divergence behavior more visible.\
> Moreover, we will include additional simulations illustrating gradient descent in the multidimensional input case for approximating polynomial target functions. These experiments will both align closely with our theoretical findings and provide a clearer, more convincing demonstration of parameter norm growth during training.\
> In particular, we will rescale the $x$-axis to demonstrate that for every activation function there exist $a,b$ such that the asymptotic trajectories of the parameters norm behave like $a \ln(x) + b$.
>
>
> - Concerning the second raised weakness: We fully acknowledge that the convergent part of Theorem 2.8 is well-known (see the reviewer's reference and [7, 3, 15, 18, 6, 35, 31] in our paper). We will make this clearer in the revision and emphasize that our main results are Theorem 3.4 and Corollary 3.6 which establish divergence under new conditions and provide novel insights into the asymptotic behavior of gradient flow.
>
> - Concerning the first question: We confirm that all identifying information and acknowledgments have been removed from the submission in accordance with the NeurIPS anonymization policy.
> In one instance, we cite a mathematical fact attributed to V. Grandjean. We included this currently unpublished fact with his permission and mentioned his name in order to respect his intellectual property. We strongly emphasize that this information cannot be used in order to identify any of the authors and that there is no conflict of interest that could possibly arise.
>
> - Concerning the second question: We agree with the reviewer and appreciate this constructive suggestion. We will include additional experiments with deterministic gradient descent, small learning rates and a large number of steps in order to match the gradient flow behavior as closely as possible.
> The preliminary simulations are consistent with the expected behavior.
>
> - We warmly thank the reviewer for reading our submission so thoroughly and for for pointing out the typos.

---

> > ### Comment · Reviewer_7NML · 2025-08-01
> >
> > Thanks for your rebuttal. I will keep the score as 5 (accept).

---

### Official Review · Reviewer_auYg · 2025-06-29

**Clarity:** 3
**Significance:** 2
**Originality:** 3
**Rating:** 5
**Confidence:** 3

**Summary:**

The submitted manuscript investigates gradient flows in the context of training fully connected neural networks. The first contribution establishes that, under a local Lipschitz continuity assumption on the gradient of the loss function, the trajectory of the gradient flow either converges to a critical point or diverges. In the divergent case, however, the value of the loss function converges to an asymptotic critical value.

The second main contribution provides a divergence result for gradient flows. Specifically, in the setting of subanalytic definable neural networks, the authors show that there exists no finite parameter configuration at which the loss achieves zero. Nonetheless, there exists a diverging sequence of parameters along which the loss converges to zero. Thus, while the global infimum of the loss is zero, it is only attained in the limit at infinity.

As a result, the authors argue that in such settings, whenever the loss at a parameter configuration is sufficiently small, the trajectory of the gradient flow necessarily diverges, yet still minimizes the loss. These theoretical findings are supported by two numerical experiments.

**Questions:**

- How does the convergence result for the gradient flow in this paper compare to other results in the literature. For example, how does it compare to that of Dello Schiavo and Maas (Local Conditions for Global Convergence of Gradient Flows and Proximal Point Sequences in Metric Spaces, Transactions of the American Mathematical Society, 2024)?
- In Appendix D, it is mentioned that the proof is partially inspired by [28]. Could you elaborate on this connection? Specifically, what aspects of the argument are derived from [28], and what constitutes the novel contribution in the presented approximation result?
- How do Theorem 3.4 and Theorem 3.5 connect to other findings of neural network approximation results?

**Ethical Concerns:**

["NO or VERY MINOR ethics concerns only"]

**Final Justification:**

The authors have addressed all of my comments satisfactorily. In particular, my primary concern - how the main results of Theorem 3.4 in neural-network approximation theory relate to the broader literature - has been fully resolved. I appreciate the manuscript’s significant contribution and am pleased to raise my score to 5 (accept).

**Limitations:**

Yes.

**Paper Formatting Concerns:**

There are no formatting concerns

**Quality:**

3

**Strengths And Weaknesses:**

**Strengths:**
- The paper extends standard convergence results to stationary points from the setting of global smoothness to one of local smoothness.
- The convergence analysis presented applies to the training of neural networks with a broad class of activation functions
- Under the SAD setting the authors, the authors establish a neural network approximation result for polynomials, showing that neural networks can approximate such functions arbitrarily well.
- By combining these results, the authors demonstrate that while the gradient flow diverges, the loss function along the trajectory converges to zero.

**Weakness:**
- The description of Theorem 2.8 at certain points slightly overstates the strength of the result. While it is true that most convergence proofs rely on global smoothness assumptions, and it is indeed valuable to see results under local smoothness, the focus here is restricted to gradient flow, i.e., the continuous-time setting. Thus, the resulting statement is expected. The continuous-time limit significantly simplifies the analysis, as it avoids the complexities associated with discrete-time methods—particularly, the need to carefully manage step sizes to ensure descent. Such considerations are inherently bypassed in the continuous-time framework, which limits the practical implications for gradient-based optimization algorithms.
- I do not fully agree with the interpretation of Figures 1 and 2. Based on the presented plots, it is not clear that the norm of the neural network parameters actually diverges; instead, they appear to approach an upper bound. This observation directly supports the concern raised in my previous point: in the discrete-time setting, there is no guarantee of monotonic decrease in the loss function, and divergence is not automatic. A more convincing evaluation would require running the optimization for more iterations to reveal any long-term trend. Additionally, adjusting the scale of the y-axis could improve the clarity and interpretability of the plots. Moreover, it would be helpful to include a plot of the gradient norm as well.

To summarize, I believe the true highlights of the paper are Theorems 3.4 and 3.5. That said, approximation theory for neural networks lies outside my primary area of expertise, so I cannot fully assess the novelty or practical impact of these results. In particular, the authors have not situated their findings within the broader literature on neural‐network approximation theory (beyond the standard universal‐approximation theorems).

---

> ### Author Rebuttal · Authors · 2025-07-30
>
> We thank the reviewer for the detailed review and suggestions for improvement. We comment on the raised points below.
>
> - Concerning the first weakness raised by the reviewer: We agree that 'the convergent part' of Theorem 2.8 is essentially known and will clarify this in the revision. Our novel contributions are Theorem 3.4 and the resulting Corollary 3.6, which we believe offer new insights into the dynamics of neural network training. While convergence analysis for SGD is generally more difficult than for gradient flows, it is important to note that in the non-convex setting, convergence to critical points is not automatic even in the gradient flow setting. In fact, there exist smooth functions for which gradient flow trajectories exhibit non-convergent behaviors, such as spiraling or limit cycles.\
> Convergence to critical points is precisely where o-minimality assumptions and Lojasiewicz-type inequalities are commonly used. Another difficulty that arises for both gradient flows and SGD is the possibility of running into one of possibly infinitely many local minima. This possibility is ruled out by our Theorem 2.8 (see item (v)) - a nontrivial implication of the o-minimality assumption.
>
>
> - Concerning the second weakness:
> The apparent plateauing of SGD in Figure 1 is due to the fact that the norm of the gradients along the gradient flow decays like $O(1/\sqrt{t})$,  where $t$ denotes training time, (see equation (7) in the paper). This implies that the parameter norms can grow at most like $O(\sqrt{t})$. Since SGD is an approximation of the gradient flow, this means that the divergence of the parameter norms is necessarily very slow and can easily be confused with a plateau. Adam’s adaptive rescaling effectively normalizes gradients to $O(1)$ - yielding an almost linear increase in the parameter norms. Therefore, we justify the use of Adam as a more effective variant of SGD, offering faster convergence in scenarios where SGD would require more iterations to achieve similar results.\
> Moreover, we will include additional simulations illustrating gradient descent in the multidimensional input case for approximating polynomial target functions. These experiments will both align closely with our theoretical findings and provide a clearer, more convincing demonstration of parameter norm growth during training.
> In particular, we will rescale the $x$-axis to demonstrate that for every activation function there exist $a,b$ such that the asymptotic trajectories of the parameters norm behave like $a \ln(x) + b$.\
> We also appreciate the suggestion to include plots of the gradient norm. These will help illustrate how the Adam optimizer mitigates gradient vanishing, producing significant updates.\
> Finally we wish to emphasize that similar sublinear‐growth of parameter norms has been observed in homogeneous models, e.g. in [49, Theorem 4.3], where gradient descent on an L-layer logistic regression model yields $\|\|\Theta(t)\|\|=O(\ln(t)^{\frac 1 L})$.
>
>
> - Concerning the first question by the reviewer: The essential distinction between our result and other convergence results lies in our *divergence proof* for gradient flows along with the introduction of a threshold $\varepsilon$ for the divergence/optimality guarantee. As pointed out above and to be emphasized in the revision, the convergent part of our Theorem 2.8 and its proof summarizes a phenomenon that is very well-understood in the literature. The reference cited by the reviewer is one such instance, offering a general convergence result under broader assumptions (non-smooth functions on arbitrary metric spaces) but significantly differs from our setting by assuming conditions that enforce boundedness of the parameters.
>
> - Concerning the second question of the reviewer:
> In the reference [28] the authors focus on the approximation of $x^2$ using only shallow neural networks (one hidden layer) and fixed activations. They demonstrate divergence for hand-constructed sequences of network parameters. These constructions are related to our general Taylor expansion argument in the proof of Theorem D.4. Our work generalizes this: we consider arbitrary polynomial target functions, deep networks, and general SAD activation functions, and we prove divergence for gradient flow trajectories, not just for manually chosen parameter sequences converging to the loss infimum.
>
>
> - Concerning the third question of the reviewer:
> We emphasize that our main contribution is Theorem 3.4 as this is, to the best of our knowledge, the first *divergence* result for neural network optimization crucially using o-minimal structures and the associated rigidity properties. Other novel contributions include the first application of definable dimension theory to neural network optimization as well as the isolation of a convenient class of activation functions (SAD activation functions) that both allow a unified mathematical analysis and includes most of the commonly used smooth activation functions. \
> We will clarify this and the relation of our work within the broader approximation theory literature in the revision.  We list some of the relevant references below. Apart from the standard universal approximation theorems which use increasing network sizes [B, C, D, E], we will also mention the surveys [F, I] and the results in [A, G, H]. The reference [A] in particular approximates polynomials in Sobolev norm with fixed-size shallow neural networks with sigmoid activation - a special case of our Theorem 3.5. In the non-smooth setting, there are moreover many results on approximation rates achieved by ReLU-networks, see [J, K, L, M, N].
>
>
> - [A] De Ryck, Tim, Samuel Lanthaler, and Siddhartha Mishra. On the approximation of functions by tanh neural networks. Neural Networks 143:732-750, 2021.
> - [B] Cybenko, George. Approximation by superpositions of a sigmoidal function. Mathematics of Control, Signals and Systems, 2(4):303–314, 1989.
> - [C] Funahashi, Ken-Ichi. On the approximate realization of continuous mappings by neural networks. Neural Networks, 2(3):183–192, 1989.
> - [D] Hornik, Kurt, Stinchcombe, Maxwell, and White, Halbert. Multilayer feedforward networks are universal approximators. Neural Networks, 2(5):359–366, 1989.
> - [E] Leshno, Moshe, Lin, Vladimir Ya., Pinkus, Allan, and Schocken, Shimon. Multilayer feedforward networks with a nonpolynomial activation function can approximate any function. Neural Networks, 6(6):861–867, 1993.
> - [F] DeVore R, Hanin B, Petrova G. Neural network approximation. Acta Numerica 30:327-444, 2021.
> - [G] Petersen, P., Raslan, M. \& Voigtlaender, F. Topological Properties of the Set of Functions Generated by Neural Networks of Fixed Size. Found Comput Math 21:375–444, 2021.
> - [H] Bölcskei, H., Grohs, P., Kutyniok, G., Petersen, P. Optimal approximation with sparsely connected deep neural networks. SIAM J. Math. Data Sci. 1(1):8–45, 2019.
> - [I] Berner J, Grohs P, Kutyniok G, Petersen P. The Modern Mathematics of Deep Learning. In: Grohs P, Kutyniok G, eds. Mathematical Aspects of Deep Learning. Cambridge University Press:1-111, 2022.
> - [J] Petersen, P., Voigtlaender, F.: Optimal approximation of piecewise smooth functions using deep ReLU neural networks. Neural Netw. 108:296–330, 2018.
> - [K] Herrmann, Lukas, Joost AA Opschoor, and Christoph Schwab. Constructive deep ReLU neural network approximation. Journal of Scientific Computing 90.2(75), 2022.
> - [L] Yarotsky, D.: Error bounds for approximations with deep ReLU networks. Neural Netw. 94:103–114, 2017.
> - [M] Daws, J., Webster, C.: Analysis of deep neural networks with quasi-optimal polynomial approximation rates, 2019. ArXiv: 1912.02302
> - [N] Opschoor, Joost AA, Philipp C. Petersen, and Christoph Schwab. Deep ReLU networks and high-order finite element methods. Analysis and Applications 18.05: 715-770, 2020.

---

> > ### Comment · Reviewer_auYg · 2025-08-01
> > **Thank you for your rebuttal**
> >
> > Thank you very much for your thorough rebuttal. You have addressed all of my comments satisfactorily. In particular, my primary concern - how the main results of Theorem 3.4 in neural-network approximation theory relate to the broader literature - has been fully resolved. I appreciate the manuscript’s significant contribution and am pleased to raise my score to 5 (accept).

---

### Official Review · Reviewer_3GA8 · 2025-07-02

**Clarity:** 2
**Significance:** 1
**Originality:** 2
**Rating:** 4
**Confidence:** 2

**Summary:**

The paper investigates gradient flows for loss landscapes of fully connected neural networks trained on polynomial targets with commonly used activation functions. The main theoretical result is a dichotomy: gradient flow either converges to a critical point or diverges to infinity while the loss converges to an asymptotic critical value, with the latter occurring for polynomial targets under mild assumptions. The key proof relies on the geometry of o-minimal structures from model theory and introduces SAD (Sublinear Analytic Definable) activation functions. Empirically, the authors run simulations on toy datasets and extend to more realistic tasks (e.g., MNIST) to demonstrate this phenomenon / main theoretical results.

**Questions:**

I am mostly curious if the theoretical analysis suggests any particular insights for the simulation results in Section 4. I'm listing several questions I have below.
- In line 292, the authors mention that Adam shows better convergence, while the SGD results in Figure 1 seem to indicate that the parameter norm plateaus. Could the authors elaborate on their justification in lines 292-298? I assume that SGD with a small learning rate is closer to the theoretical analysis (gradient flow) than Adam updates.
- Does this behavior still occur with extremely poor initialization or with other activation functions (e.g., Relu or even identity)? I speculate that they will show the same behavior as in Figure 2. If so, what is the practical significance of the proposed work?

**Ethical Concerns:**

["NO or VERY MINOR ethics concerns only"]

**Final Justification:**

The authors have convincingly addressed my questions (e.g., it makes sense that the parameter growth happens when we have scale invariant models such as with batch norm.)

**Limitations:**

Yes.

**Quality:**

2

**Strengths And Weaknesses:**

Note: I don’t have the necessary background to evaluate the correctness/significance of this work, hence my confidence score is 2. I will communicate with other reviewers to address any misunderstanding.

Strengths:
- The paper is generally well-written, and the source code is provided for reproducibility.
- A detailed experimental setup is provided in Appendix E.
- While I cannot comment on the correctness/significance of the theoretical derivations, the derivations and proofs are well organized (e.g., Figure 4).

Weaknesses:
- The empirical results don't convincingly demonstrate that the theoretical findings translate into practical settings. For example, it is well known that parameter norms continue to increase during training (motivating techniques like L2 regularization), making it difficult to understand whether the theoretical insights are specifically captured in Figure 1. Does this phenomenon fail to occur with extremely poor initialization or with activation functions such as ReLU (as predicted by theory)? Please see the questions section below.
- Similar concerns apply to the empirical results presented in Section 4.2. At the moment, the theoretical analysis does not appear to capture any important practical considerations that aren't already well-understood.
- The theoretical results apply only to gradient flows and smooth activation functions, which significantly reduces the practical relevance.

Minor:
- It would be helpful to have detailed related work, e.g., other applications of o-minimality to neural networks.

---

> ### Author Rebuttal · Authors · 2025-07-30
>
> The authors thank the reviewer for the detailed and constructive feedback.
> Below, we address each of your questions in turn.
>
> - To directly answer the first question: SGD with a sufficiently small learning rate indeed closely follows the continuous gradient flow, while Adam’s adaptive rescaling obscures the slow divergence predicted by our theory.
> The impression that SGD plateaus is due to the fact that the norms of the gradients along the gradient flow decay like $O(1/\sqrt{t})$, where $t$ denotes training time, which follows from equation (7). This implies that the parameter norms can grow at most like $O(\sqrt{t})$ and therefore look like they are plateauing although they are slowly diverging to infinity.
> Adam’s second order momentum‐based normalization effectively re-scales each gradient coordinate to $O(1)$. Therefore, we justify the use of Adam as a more effective variant of SGD, offering faster convergence in scenarios where SGD would require more iterations to achieve similar results.\
> Moreover, we will include new simulations illustrating gradient descent in the multidimensional input case for approximating polynomial target functions. These experiments will both align closely with our theoretical findings and provide a clearer, more convincing demonstration of parameter norm growth during training.
> In particular, we will rescale the $x$-axis to demonstrate that for every activation function there exist $a,b$ such that the asymptotic trajectories of the parameters norm behave like $a \ln(x) + b$.\
> Finally we wish to emphasize that similar sublinear‐growth of parameter norms has been observed in homogeneous models, e.g. in [49, Theorem 4.3], where gradient descent on an L-layer logistic regression model yields $\|\Theta(t)\|=O(\ln(t)^{\frac 1 L})$.
>
> - Concerning the second question: We appreciate the reviewer's interest in how the experiments would look with other activations or extremely poor initialization.
> First, regarding poor initialization, we found that in our setting, SGD rarely gets stuck in local minima purely due to initialization, owing to the simplicity of the model and loss landscape. While it is possible to force standard gradient descent into local traps through carefully chosen initializations, we observed that using a sufficiently large dataset effectively prevents such issues by smoothing the loss surface and guiding the optimization away from problematic regions. Of course, in more complex architectures or highly non-convex settings, initialization may have a stronger effect, but in our setting, we did not observe significant sensitivity.\
> Regarding other activation functions, we verified the same phenomenon with the SiLU, which satisfies our SAD conditions. In contrast, with the identity activation, the network fails to converge to the target function, and we observed a decrease in the parameter norm during training. Note however, that we excluded polynomial activation functions from Theorem 3.5 and Corollary 3.6.
> Although ReLU falls outside our SAD framework, prior work on homogeneous ReLU networks [49] proves similar divergence behaviors, and our preliminary experiments confirm divergent trajectories.\
> Finally, we note that unbounded parameter‐norm growth is a relatively specialized phenomenon. In many practical settings involving non-homogeneous activations or more complex loss landscapes, parameter norms tend to fluctuate during training and stabilize over time. In the literature, pronounced divergence has been documented primarily in scale‑invariant networks, where scaling the weights leaves the model’s predictions unchanged (only their magnitude shifts).
>
> Concerning the weaknesses, we emphasize that our main contributions are the advance in the mathematical tools used in the theoretical analysis of machine learning and the understanding of training dynamics of neural networks.\
> Moreover, we recognize that results for SGD would be more directly applicable than results for gradient flows. We will pursue these in future work. We do however believe that the gradient flow setting is an important proof of concept and resembles the gradient descent setting with a small step-size. The treated smooth activation functions however are highly relevant in practice. In particular the smooth GELU function which we cover is frequently used in scientific computing problems as well as in Transformer models like BERT and GPT-3.
>
> Furthermore, we agree that it would be helpful to have more detailed work on the applications of o-minimality to neural networks. The upshot is that the *convergence analysis* for definable objective functions, under the assumption that the parameters stay bounded, is very well understood (see for instance references [7, 3, 15, 18, 6, 35, 31] in the paper and [A, B] below). In particular, the convergent part of our Theorem 2.8 is essentially well-known, a fact that we will emphasize in the revision. Other applications of o-minimality include alignment and directional convergence [34, 42, 49] or bounds on the VC-dimension [38, 39]. The significant contribution of our paper is that it is, to the best of our knowledge, the first paper using o-minimal structures to achieve *divergence* results in the training of neural networks. This is the content of our Theorem 3.4 and our main result Corollary 3.6. Another key contribution is that our definition of *SAD* activation functions provides a convenient mathematical framework that simplifies and conceptualizes many arguments while covering most smooth activation functions used in practice.
>
> - [A] Dello Schiavo Lorenzo, Jan Maas, and Francesco Pedrotti. "Local conditions for global convergence of gradient flows and proximal point sequences in metric spaces." Transactions of the American Mathematical Society 377(6):3779-3804, 2024.
> - [B] Hedy Attouch, Jérôme Bolte, Patrick Redont, and Antoine Soubeyran. "Proximal alternating minimization and projection methods for nonconvex problems: An approach based on the Kurdyka-Łojasiewicz inequality." Mathematics of Operations Research, 35(2):438–457, 2010.

---

> > ### Comment · Reviewer_3GA8 · 2025-08-02
> >
> > Thank you for addressing the questions. I raised my score to 4.

---

### Author Response · Authors · 2025-08-08
**Authors’ Response to Reviewers**

We thank all the reviewers for all the insightful feedback that led to improvements in the paper.  We greatly appreciate the engagement and the re-evaluation.

---

### Decision · Program_Chairs · 2025-09-17

**Decision:**

Accept (poster)

**Comment:**

This is a mainly theoretical work, culminating in a proof of divergence for gradient flows for deep neural networks with analytic activation and polynomial target functions.  The techniques used are non-trivial, based on geometry of o-minimal structures.

Reviewers praised the quality of the writing and the generality of the approach.

On the other hand, they had concerns about distinctions from prior work, the restriction to smooth activation functions, relevance of the numerical experiments, and provision of helpful proof sketches.

The authors provided a detailed rebuttal, which was followed by some discussion with the reviewers.  Several reviewers consequently raised their ratings.

All reviewers recommend or lean towards acceptance, which I also support.  The authors should ensure that they address all the comments when revising the paper.